# Probabilistic Neural Pruning via Sparsity Evolutionary Fokker-Planck-Kolmogorov Equation

**Zhanfeng Mo**[†]  **Haosen Shi**[‡]  **Sinno Jialin Pan**[†‡]

[†] Nanyang Technological University, Singapore  [‡] The Chinese University of Hong Kong

zhanfeng001@ntu.edu.sg; haosen.shi.ai@link.cuhk.edu.hk; sinnopan@cuhk.edu.hk

## Abstract

Neural pruning aims to compress and accelerate deep neural networks by identifying the optimal subnetwork within a specified sparsity budget. In this work, we study how to gradually sparsify the unpruned dense model to the target sparsity level with minimal performance drop. Specifically, we analyze the evolution of the population of optimal subnetworks under continuous sparsity increments from a thermodynamic perspective. We first reformulate neural pruning as an expected loss minimization problem over the mask distributions. Then, we establish an effective approximation for the sparsity evolution of the optimal mask distribution, termed the **S**parsity Evolutionary **F**okker-**P**lanck-**K**olmogorov Equation (**SFPK**), which provides closed-form, mathematically tractable guidance on distributional transitions for minimizing the expected loss under an infinitesimal sparsity increment. On top of that, we propose SFPK-pruner, a particle simulation-based probabilistic pruning method, to sample performant masks with desired sparsity from the destination distribution of SFPK. In theory, we establish the convergence guarantee for the proposed SFPK-pruner. Our SFPK-pruner exhibits competitive performance in various pruning scenarios. The code is available on GitHub[1].

## 1 Introduction

Overparameterized deep neural networks have shown remarkable success across diverse applications (He et al., 2016; Howard et al., 2017; Redmon et al., 2016; Radosavovic et al., 2020; Dosovitskiy et al., 2021; Touvron et al., 2021; Devlin et al., 2018; Zhang et al., 2015; Brown et al., 2020). However, their considerable parameter size significantly hinders their efficiency, posing challenges on edge computing scenarios (Wu et al., 2016; Chen & Ran, 2019; Yao et al., 2017; Han et al., 2017; Bhattacharya & Lane, 2016). To mitigate this issue, a variety of model compression methods (Han et al., 2015; He et al., 2018; Hinton et al., 2015; Polino et al., 2018; Courbariaux et al., 2015; Chen et al., 2019) have been established to reduce the model size and computation cost with minimal performance drop. Neural pruning is one of the most mainstream methods due to its practicality and effectiveness (Lee et al., 2019; Tanaka et al., 2020; Louizos et al., 2018; Chen et al., 2021; Zhuang et al., 2020; Frankle & Carbin, 2019; Zhu & Gupta, 2018a). The objective of neural pruning is to remove unimportant parameters of a dense model until a specified sparsity constraint is met while minimizing the model loss to the greatest extent possible. In general, neural pruning is challenging, as it requires identifying optimal subnetworks under a sparsity constraint, which is a high-dimensional zero-one programming problem (Papadimitriou & Steiglitz, 1998).

*Can we mitigate the difficulty of neural pruning if a set of performant subnetworks with slightly lower sparsity is readily accessible?* In practice, a performant subnetwork can be slightly sparsified with only a small performance drop through minor and careful modifications (Hu et al., 2016; Srinivas & Babu, 2015). This implies the potential to derive the optimal subnetworks at the desired sparsity level by making minor adjustments to slightly denser optimal subnetworks. Thus, we are motivated to study how the population of optimal subnetworks evolves when there is a small increase in sparsity. Following this sparsity evolution of the optimal subnetwork population, the

---

[1]https://github.com/mzf666/SFPK-main

unpruned model, which initially consists of the optimal subnetwork population with zero sparsity, can be gradually transformed into the desired optimal subnetworks at the target sparsity level. As a result, we can prune the model by sampling performant subnetworks from this final population.

*How does the population of optimal subnetworks change as sparsity increases progressively?* An initial attempt to address this question was made in (Mo et al., 2023), which aims to approximate the trajectory of a carefully constructed optimal subnetwork as sparsity increased from zero to one. However, this approach only considers a single optimal subnetwork at each sparsity level, failing to capture the entire population. Furthermore, it assumes that a global optimum could always be found within the vicinity of an arbitrary optimal subnetwork, which is overly restrictive in practice.

Different from (Mo et al., 2023), our approach draws inspiration from thermodynamics, which studies the evolution of particle populations through time-dependent particle density functions. These functions describe the probability distribution of the particle coordinate or velocities under random forces (Kadanoff, 2000). Following this spirit, we study the sparsity evolution of the population of optimal subnetworks from a probabilistic perspective. At each sparsity level, we apply the probabilistic lifting technique (Wild et al., 2023) to redefine neural pruning as a problem of expected loss minimization over the distribution space of subnetworks. From the thermodynamic perspective, each subnetwork can be analogized to a particle, while a subnetwork distribution at a specific sparsity level represents a particle density pattern at a particular time. In this context, the expected loss can be considered as the overall energy of the particle system. The optimal subnetwork distribution that minimizes the expected loss effectively summarizes the quality of the entire optimal subnetwork population by assigning a high weight to each optimum. Therefore, we can approximate the sparsity evolution of the population of optimal subnetworks by tracking the sparsity evolution of the optimal subnetwork distribution. Instead of solving the optimal subnetwork distribution from scratch at the target sparsity level, this approach breaks down the process into a series of tractable transitions between optimal subnetwork distributions at gradually increasing sparsity levels. As demonstrated later, each transition step can be achieved by carefully refining the quality assignments of the microscopic subnetworks over the support of the current sparsity level's optimal distribution.

**Contributions.** 1) We establish a novel probabilistic pruning framework, termed the **S**parsity Evolutionary **F**okker-**P**lanck-**K**olmogorov Equation (**SFPK**) (Section 4.2): conceptually similar to the traditional Fokker-Planck-Kolmogorov (FPK) equation, which models the time evolution of particle density in thermodynamics, our proposed SFPK theory offers a closed-form, mathematically tractable approach to approximate the evolution of the optimal subnetwork distribution as sparsity gradually increases, serving as an analog to the FPK equation in probabilistic neural pruning. 2) we develop SFPK-pruner, a principled pruning method based on particle simulation, which utilizes SFPK to effectively sample performant subnetworks with desired sparsity (Section 4.3); additionally, we prove the convergence of the SFPK-pruner in Proposition 3; 3) we conduct numerical experiments showing that SFPK-pruner achieves competitive performance across various models and datasets in both structured and unstructured pruning scenarios (Section 5).

## 2 RELATED WORK

Various pruning frameworks have been proposed. Score-based methods prune parameters based on their importance scores. Different metrics (Han et al., 2015; Tanaka et al., 2020; Wang et al., 2020; Lubana & Dick, 2021; Lee et al., 2019; Jacot et al., 2018; Rachwan et al., 2022) are proposed as the score function to determine which weights to prune. However, such an estimation is shortsighted for pruning a lot of parameters at once. Regularization-based methods (Louizos et al., 2018; Chen et al., 2021; Zhuang et al., 2020) prune by converting the original binary constrained optimization problem into a continuous unconstrained optimization problem and incorporating soft sparsity penalties. However, these penalties are numerically unstable, making it difficult to converge to a desired sparsity level. Sparse training methods (Zhu & Gupta, 2018a; Frankle & Carbin, 2019; Morcos et al., 2019; Tai et al., 2022) iteratively prune and fine-tune to approximate optimal sparse subnetworks, yet the computational demands of jointly optimizing parameters and subnetworks limit their scalability to large models and datasets.

Bayesian-based neural pruning approaches typically incorporate the distribution of model parameters while imposing specific sparsity constraints. In (Neklyudov et al., 2017; Louizos et al., 2017), certain priors are utilized to push the posterior distribution of the model parameters to a sparse distri-

bution, facilitating subsequent pruning. In (Zhao et al., 2019; Molchanov et al., 2017a), a Bayesian neural network method called variational dropout is used, where unimportant connections are discarded based on the dropout results. From a high-level perspective, these methods share similarities with our approach in terms of modeling the behavior of the mask distribution. However, there are two distinctions: 1) we focus on a series of marginal distributions under different sparsity levels rather than the entire parameter distribution spaces, and 2) we do not follow the posterior estimation paradigm, but instead use a stochastic differential equation to model the evolution of the distribution.

## 3 PRELIMINARIES

**Basic Notations.** In this paper, the dense neural network to be pruned is parameterized by $\boldsymbol{\theta} \in \boldsymbol{\Theta} \subset \mathbb{R}^d$, with $\boldsymbol{\Theta}$ representing the parameter space of the $d$ prunable parameters. The $\ell_p$-norm of a vector $\mathbf{v}$ is defined as $\|\mathbf{v}\|_p \triangleq (\sum_i |\mathbf{v}[i]|^p)^{1/p}$, where $\mathbf{v}[i]$ denotes the $\mathbf{v}$'s $i$-th entry. The element-wise product is denoted by $\odot$. $[n] \triangleq \{1, ..., n\}$. $B_r \triangleq \{\mathbf{m} \in \mathbb{R}^d : \|\mathbf{m}\|_2 \leqslant r\}$ denotes the Euclidean ball with radius $r > 0$. We use the terms "probability measure" and "distribution" interchangeably without specification. For any measurable set $M \subset \mathbb{R}^n$, $\mathcal{P}(M)$ defines the collection of all probability measures with a density function (w.r.t Lebesgue measure) supported on $M$. $\text{Dirac}(\cdot; \mathbf{m})$ denotes the Dirac delta distribution concentrated on $\mathbf{m}$. $\text{supp}(\pi)$ denotes the support of the distribution $\pi$. $\text{Unif}(M)$ denotes the uniform distribution over $M$. $\nu(\cdot|\cdot) : \mathbb{R}^d \times \mathbb{R}^d \mapsto \mathbb{R}_+$ denotes a probability kernel, where $\nu(\cdot|\mathbf{m}) \in \mathcal{P}(\mathbb{R}^d)$, and the mapping $\mathbf{m} \mapsto \nu(M|\mathbf{m})$ is measurable for any measurable $M \subset \mathbb{R}^d$. Thus, $\nu(\cdot|\mathbf{m})$ represents a distribution conditioned on $\mathbf{m}$. $\nu * \pi \triangleq \int \pi(\mathbf{m} - \boldsymbol{\delta})\nu(\boldsymbol{\delta}|\mathbf{m} - \boldsymbol{\delta})\mathrm{d}\boldsymbol{\delta}$ represents the convolution between a probability kernel $\nu$ and a distribution $\pi$. $\text{law}(\mathbf{m})$ denotes the distribution of the random variable $\mathbf{m}$.

**Neural Pruning.** In neural pruning, given a dense neural network $\boldsymbol{\theta}^*$ and a parameter budget $d'$ (typically $d' \ll d$), the objective is to identify the optimal subnetwork containing at most $d'$ non-zero parameters while maximizing the model performance: $\min_{\mathbf{m} \in \{0,1\}^d} L(\mathbf{m} \odot \boldsymbol{\theta}^*)$, s.t. $\|\mathbf{m}\|_0 = d'$. Here, $L(\cdot) : \mathbb{R}^d \mapsto \mathbb{R}^+$ is a model energy function to be minimized (such as evaluation loss, predictive error, or generalization bound). $\mathbf{m}$ is a binary mask, where $\mathbf{m}[i]$ equals to 0 indicates that the $i$-th parameter is pruned, and vice versa. We refer to $1 - d'/d$ as the target sparsity of neural pruning. The population of optimal masks refers to the set of masks that achieve the minimum model energy.

**Probabilistic Lifting and Convexification.** The probabilistic lifting and convexification technique, initially devised to address the notorious nonconvexity in optimization problems (Wild et al., 2023, Section 2), involves two key steps: 1) lifting the original Euclidean nonconvex optimization problem to the distribution space, and 2) incorporating a convex regularizer, as outlined in (1).

$$\min_{x \in \Omega} l(x) \stackrel{\text{Probabilistic lifting}}{\Longrightarrow} \min_{\pi \in \mathcal{P}(\Omega)} \langle l, \pi \rangle \triangleq \int_\Omega l(x)\pi(\mathrm{d}x) \stackrel{\text{Convexification}}{\Longrightarrow} \min_{\pi \in \mathcal{P}(\Omega)} \langle l, \pi \rangle + \lambda \mathcal{R}[\pi], \quad (1)$$

where $l(\cdot)$ denotes the nonconvex objective function, $\Omega$ denotes the nonconvex feasible region, $\mathcal{R}[\cdot]$ denotes a convex functional, and $\lambda > 0$ is the regularization rate. This approach revises the original optimization into a convex distributional optimization problem by substituting $\Omega$ with the convex distribution space $\mathcal{P}(\Omega)$, and replacing $l(\cdot)$ with the convex functional $\langle l, \cdot \rangle + \mathcal{R}[\cdot]$. Thanks to the introduced convexity, the global optimum of (1) is proven to uniquely exist (Wild et al., 2023). Furthermore, working within the distribution space provides greater convenience for conducting continuous evolution analysis and convergence analysis. Admittedly, these conveniences come with trade-offs: 1) The finite-dimensional optimization problem is lifted to the infinite-dimensional distribution space, which is numerically more challenging to address. 2) The convex regularizer introduces a gap proportional to $\lambda$ between the initial objective and the convexified one. To identify the optimum of (1), one needs to resort to generalized variational inference methods (Knoblauch et al., 2022), such as optimizing over a family of finite-dimensional parameterized distributions or running infinite-dimensional gradient flow in the distribution space (Louizos & Welling, 2017; Wild et al., 2024; Santana et al., 2021; Glaser et al., 2021; D'Angelo & Fortuin, 2021).

**Sparsity Evolution of the Optimal Mask Distribution.** With the probabilistic lifting and convexification technique, we can feel free to analyze the sparsity evolution of the population of optimal masks in a probabilistic pruning scenario. Let $\pi_t$ be the optimal mask distribution that minimizes the probabilistic pruning objective at sparsity level $t$. In intuition, $\pi_t$ weights the quality of each mask

of sparsity $t$: it assigns large weights to highly performant masks while assigning minimal, or even zero, weights to poorly performing masks. Thus, the dynamic of $t \mapsto \pi_t$ is essentially characterized by the evolution of the assignment of mask quality under progressive sparsity increments. In practice, a performant mask can be slightly sparsified with a small performance drop through minor and careful alterations. This suggests that, for a small $\Delta t$-sparsity increment, one can derive $\pi_{t+\Delta t}$ from $\pi_t$ through minor adjustments on the quality assignment scheme of $\pi_t$. If $\pi_t$ is readily accessible, deriving a local transition from $\pi_t$ to $\pi_{t+\Delta t}$ would be easier than solving $\pi_{t+\Delta t}$ from scratch. Suppose the transition from $\pi_t$ to $\pi_{t+\Delta t}$ can be formulated as $\mathcal{T}[\pi_t]$ [2]. By informally taking $\Delta t \to 0$, the infinitesimal evolution of $\pi_t$ can be derived as a distribution-valued differential equation:

$$\pi_{t+\Delta t} - \pi_t = \mathcal{T}[\pi_t]\Delta t \quad \overset{\Delta t \to 0}{\Longrightarrow} \quad \partial_t \pi_t = \mathcal{T}[\pi_t], \quad t \in [0, 1 - d'/d]. \tag{2}$$

Note that the formulation of (2) bears a great similarity with the standard FPK equation (Bogachev et al., 2015), which takes the form $\partial_t p_t = \mathcal{A}^*[p_t]$. Here, $p_t$ is the particle density at time $t$, and $\mathcal{A}^*[\cdot]$ is a differential operator determined by the microscopic particle dynamic. While the standard FPK equation describes the time evolution of the particle density, (2) describes the distributional transition $\pi_t$ under an $dt$-sparsity increment, making it the sparsity evolutionary analog of the thermodynamic FPK equation. Therefore, we designate (2) as the Sparsity Evolutionary Fokker-Planck-Kolmogorov Equation (SFPK). SFPK provides explicit guidance on transforming $\pi_0$ (the initial optimal mask distribution which is highly concentrated on the unpruned model) to $\pi_{1-d'/d}$, which enables us to sample performant masks within the parameter budget $d'$. Next, we focus on formalizing the derivation of the SFPK and developing the SFPK-guided probabilistic pruning algorithm.

## 4 METHODOLOGY

In this section, we first introduce Probabilistic Soft Neural Pruning (Definition 1), a convex probabilistic relaxation that facilitates our derivation, with provable uniqueness and the existence of its global optimal mask distribution at any sparsity level (Proposition 1). On top of that, we derive SFPK as an approximation of the sparsity evolution of the optimal mask distributions (Proposition 2 and Proposition 3). Finally, we propose SFPK-Pruner, a particle simulation-based pruning method (Algorithm 1) that enables us to sample performant masks from the destination distribution of SFPK.

### 4.1 PROBABILISTIC SOFT NEURAL PRUNING

There are two research issues for formally establishing SFPK following the derivations in (2). Firstly, the minimal increment of the $\ell_0$-norm based discrete sparsity is $1/d$, which hinders us from taking the limit $\Delta t \to 0$. Secondly, the optimal mask distribution undergoes drastic fluctuations even under an infinitesimal sparsity increase, as there is no overlap between the sets of masks at different sparsity levels. This leads to a spiky change in the support of $\pi_t$, preventing us from analyzing the continuous transition from $\pi_t$ to $\pi_{t+dt}$.

To address these two issues, we first generalize the discrete sparsity to continuous values. We then relax the sparsity constraint to ensure smoother transitions in mask distributions as sparsity increases. This yields the **relaxed soft pruning** problem:

$$\min_{\mathbf{m} \in M_t^\rho} L_\varepsilon(\mathbf{m}) \triangleq L(P_\varepsilon(\mathbf{m}) \odot \boldsymbol{\theta}^*), \quad M_t^\rho \triangleq \left\{ \mathbf{m} \in \mathbb{R}^d : s(\mathbf{m}) \in [t, t+\rho] \right\}. \tag{3}$$

Here, $\boldsymbol{\theta}^*$ is the dense model, $t \in [0, 1]$ is the target sparsity, $s(\cdot) : \mathbb{R}^d \mapsto [0, 1]$ is a soften sparsity function (for example, $1 - \|\cdot\|_p^p/d$) that satisfies $s(\mathbf{m}) = 1 - \|\mathbf{m}\|_0/d$, for any $\mathbf{m} \in \{0, 1\}^d$. To ensure that the model loss is evaluated on sparse masks, we polarize the soft mask to be $\varepsilon$-nearly binary using the mask polarizer $P_\varepsilon$ proposed in (Mo et al., 2023, Appendix A.3) [3]. Thereby, an infinitesimal change of the softened sparsity becomes well-defined, while the inherent sparsity of neural pruning is preserved. As $\varepsilon \to 0$ and $\rho \to 0$, (3) recovers the original neural pruning problem, making it an amenable proxy of neural pruning. In the sequel, the term "sparsity" refers to the soft sparsity $s(\cdot)$ without further specification. We then augment the relaxed soft pruning problem through probabilistic lifting and convexification.

---

[2] The formal expression of the distributional transition $\partial_t \pi_t = \mathcal{T}[\pi_t]$ will be introduced in Proposition 2.

[3] Let $I_\varepsilon \triangleq ([0, \varepsilon] \cup [1 - \varepsilon, 1])^d$, we set $P_\varepsilon(\mathbf{m}) \triangleq \arg\min_{\mathbf{m}' \in I_\varepsilon \cap M_t^\rho} \|\mathbf{m}' - \mathbf{m}\|_2$.

**Definition 1** (Probabilistic Soft Neural Pruning). Following the previously introduced notations, the **Probabilistic Soft Neural Pruning** problem at sparsity level $t$ is defined as follows:

$$\min_{\pi \in \mathcal{P}(M_t^\rho)} \langle L_\varepsilon, \pi \rangle + \lambda \mathcal{R}[\pi] \triangleq \int_{M_t^\rho} L_\varepsilon(\mathbf{m})\pi(\mathrm{d}\mathbf{m}) + \frac{\lambda}{2} \int_{M_t^\rho} \kappa(\mathbf{m}, \mathbf{m}')\pi(\mathrm{d}\mathbf{m})\pi(\mathrm{d}\mathbf{m}'), \quad (4)$$

where $\lambda > 0$ is the regularization rate and $\kappa(\cdot, \cdot) : \mathbb{R}^d \times \mathbb{R}^d \mapsto \mathbb{R}_+$ is a convex smooth kernel. A practical choice of $\kappa$ is the radial basis function kernel (Gretton et al., 2012), i.e. $\kappa(\mathbf{m}, \mathbf{m}') \triangleq \exp(-\|\mathbf{m} - \mathbf{m}'\|_2^2/d)$.

*Remark* 1. (4) possess a thermodynamics interpretation: each mask $\mathbf{m}$ represents the particle coordinate, $\pi$ represents the particle location density, $\langle L_\varepsilon, \pi \rangle$ quantifies the external potential that acts on individual particles, and $\mathcal{R}[\pi]$ measures the pair-wise interaction energy, such as repulsive energy. From this perspective, neural pruning is equivalent to identifying the equilibrium of the particle system that minimizes the overall energy of the system. However, since the mask sparsity is restricted to $[t, t + \rho]$, (4) is a constrained thermodynamics system.

As noted in (Arbel et al., 2019, Lemma 25), the convexity of the kernel regularizer $\kappa(\cdot, \cdot)$ implies the convexity of $\mathcal{R}[\cdot]$. Unlike the relaxed soft pruning (3), which aims to find the optimal masks of the nonconvex objective $L_\varepsilon(\cdot)$ at sparsity level $t$, (4) aims to identify the distributional minimizer of the convex functional $\langle L_\varepsilon, \cdot \rangle + \lambda \mathcal{R}[\cdot]$ within the sparsity range $[t, t + \rho]$. As $\lambda$ and $\rho$ tends to 0, (4) recovers the original polarized soft neural pruning, in the sense that any distribution over the mixture of optimal masks that minimizes (3) is an optimum of (4). Therefore, (4) is indeed a probabilistic extension of the relaxed soft pruning problem while inheriting its theoretical merits in continuous sparsity analysis. Moreover, (4) benefits from the provable existence and uniqueness of its global minimizer, as demonstrated in Proposition 1. The proof is detailed in Appendix D.1.

**Proposition 1** (Existence and Uniqueness of the Optimal Mask Distribution (informal version of Proposition 4)). *Under some regularity conditions on model loss $L_\varepsilon(\cdot)$, the global minimizer of probabilistic soft neural pruning (4) uniquely exists. This global minimizer is denoted as $\pi_t$, and referred to as the "**optimal mask distribution at sparsity level** $t$".*

## 4.2 Sparsity Evolutionary Fokker-Planck-Kolmogorov Equation (SFPK)

Existing distributional gradient flow methods (Glaser et al., 2021; D'Angelo & Fortuin, 2021; Wild et al., 2023) are primarily designed for unconstrained optimization problems and are not effective in solving the probabilistic soft neural pruning problem (4) within the feasible sparsity range of $[t, t+\rho]$. Therefore, an alternative approach is necessary to solve $\pi_{1-d'/d}$ in a more tractable manner. As discussed in Section 1, we aim to approximate the evolution of optimal mask distribution $t \mapsto \pi_t$ as the sparsity $t$ increases from 0 to $1 - d'/d$, following SFPK introduced in (2). By gradually transforming $\pi_0$ (which can be well approximated by the Dirac distribution concentrated at the unpruned mask), we can achieve the desired $\pi_{1-d'/d}$.

The derivation of SFPK in (2) hinges on determining the optimal distributional transition operator $\mathcal{T}[\cdot] : \mathcal{P}(\mathbb{R}^d) \mapsto \mathcal{P}(\mathbb{R}^d)$. In essence, $\mathcal{T}[\pi_t]$ is expected to push the support of $\pi_t$ to be $\mathrm{d}t$-sparser while ensuring a minimal increase in the expected pruning objective. To obtain $\mathcal{T}[\pi_t]$, we need to estimate $\pi_{t+\Delta t}$ based on $\pi_t$. To this end, we employ a **distributional localization** technique, which involves two steps: 1) localizing the feasible region to a vicinity of $\pi_t$, and 2) linearizing the functional objective of (4) at $\pi_t$. Specifically, we consider candidate distributions that are reachable from $\pi_t$ by a local transition:

$$\widehat{\mathcal{P}}\left(M_{t+\Delta t}^\rho\right) \triangleq \{\nu * \pi_t \mid \mathrm{supp}(\nu(\cdot|\mathbf{m})) \in B_{r_t \Delta t}, \ \forall \ \mathbf{m} \in M_t^\rho\} \cap \mathcal{P}\left(M_{t+\Delta t}^\rho\right). \quad (5)$$

Here, $r_t > 0$ is the localization radius, and $\nu(\cdot|\cdot)$ is a transition kernel, where $\nu(\cdot|\mathbf{m})$ denotes a transition distribution conditioned on $\mathbf{m}$. The convolution $\nu * \pi_t$ indicates that each candidate distribution can be instantiated by $\mathbf{m}_t + \boldsymbol{\delta}$, where $\mathbf{m}_t \sim \pi_t$ is the optimal probabilistic mask and $\boldsymbol{\delta} \sim \nu(\cdot|\mathbf{m}_t)$ is a local transition conditioned on $\mathbf{m}_t$. The constraint in (5) ensures that $\boldsymbol{\delta}$ is a minor displacement with $\|\boldsymbol{\delta}\|_2 \leqslant r_t \Delta t$. The intersection in (5) implies that $\boldsymbol{\delta}$ pushes the mask from $M_t^\rho$ to $M_{t+\Delta t}^\rho$, resulting in a $\Delta t$-sparsity increment. Additionally, based on the functional first-order expansion (Ernzerhof, 1994), the probabilistic pruning objective $\mathcal{L}[\cdot] \triangleq \langle L_\varepsilon, \cdot \rangle + \lambda \mathcal{R}[\cdot]$ can be linearized at $\pi_t$ as:

$$\mathcal{L}[\pi] \approx \mathcal{L}[\pi_t] + \left\langle \frac{\delta}{\delta \pi_t} \mathcal{L}, \pi - \pi_t \right\rangle = \mathcal{L}[\pi_t] + \left\langle L_\varepsilon + \lambda \mu_{\pi_t}, \pi - \pi_t \right\rangle, \quad (6)$$

**Algorithm 1** SFPK-pruner

1: **Input:** target model $\boldsymbol{\theta}^*$, target budget $d'$, localization radius $r_t$, particle number $n$, simulation steps $K$.
2: **Output:** a performant binary mask $\widehat{\mathbf{m}}$.
3: $t \leftarrow 0, \Delta t \leftarrow (1 - d'/d)/K$.
4: Initialize $\mathbf{m}_t^i \leftarrow \mathbf{1}, i = 1, ..., n$.
5: **for** $k = 1$ **to** $K$ **do**
6:    $\widehat{\pi}_t \leftarrow \mathrm{Unif}(\{\mathbf{m}_i^t\}_{i=1}^n)$
7:    **for** $i = 1$ **to** $n$ **do**
8:       $\mathbf{m}_{t+\Delta t}^i \leftarrow \mathbf{m}_t^i + T(\mathbf{m}_t^i; \widehat{\pi}_t)\Delta t$
9:    **end for**
10:    $t \leftarrow t + \Delta t$
11: **end for**
12: $\widehat{\mathbf{m}} \leftarrow \mathrm{top}_{d'}\left(\sum_{i=1}^n \mathbf{m}_t^i\right)$
13: **return** $\widehat{\mathbf{m}}$

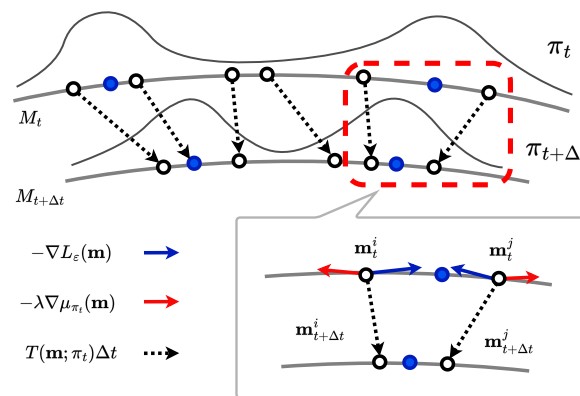

Figure 1: Illustration of the optimal local transition. Each white dot denotes a mask particle, and each blue dot denotes the local optimum of $L_\varepsilon$. The black curves denote the density function of $\pi_t$ and $\pi_{t+\Delta t}$ that are highly concentrated on the blue optima.

where $\frac{\delta}{\delta \pi_t}\mathcal{L}$ is the functional derivative of $\mathcal{L}$ at $\pi_t$, and $\mu_{\pi_t}(\cdot) \triangleq \int \kappa(\cdot, \mathbf{m})\pi_t(\mathrm{d}\mathbf{m})$ is the $\kappa$-kernel mean embedding of $\pi_t$. We now introduce the one-step distributional transition problem as a localized approximation of (4).

**Definition 2** (Local Distributional Transition). Using the previously introduced notations, the Local Distributional Transition problem associated with $\pi_t$ is defined as

$$\min_\nu \left\langle L_\varepsilon + \lambda\mu_{\pi_t}, \nu * \pi_t - \pi_t \right\rangle, \text{ s.t. } \nu * \pi_t \in \widehat{\mathcal{P}}\left(M_{t+\Delta t}^\rho\right). \tag{7}$$

We refer to the minimizer of (7), denoted by $\nu_t$, as the **optimal local transition kernel**. The following proposition establishes an approximation for $\nu_t$. The proof is detailed in Appendix D.1.

**Proposition 2** (Optimal Local Distributional Transition (informal version of Proposition 5 presented in Appendix)). *Under some regularity conditions on $L_\varepsilon(\cdot)$ and $s(\cdot)$, the optimum of (7) is attained at $\nu_t(\cdot|\mathbf{m}) \triangleq \mathrm{Dirac}(\cdot; T(\mathbf{m}; \pi_t)\Delta t)$ within an error tolerance of order $\mathcal{O}(\Delta t^2)$. Let $\mathbf{g} \triangleq -\nabla L_\varepsilon(\mathbf{m}) - \lambda\nabla\mu_{\pi_t}(\mathbf{m})$, $\mathbf{s} \triangleq \nabla s(\mathbf{m})$, and $r_t\|\mathbf{s}\|_2 > 1$, $T(\mathbf{m}; \pi_t)$ is given by*

$$T(\mathbf{m}; \pi_t) \triangleq \frac{\mathbf{s}}{\|\mathbf{s}\|_2^2} + \left(\frac{r_t^2\|\mathbf{s}\|_2^2 - 1}{\|\mathbf{s}\|_2^2\|\mathbf{g}\|_2^2 - (\mathbf{s}^\top\mathbf{g})^2}\right)^{\frac{1}{2}} \cdot \left(\mathbf{I} - \frac{\mathbf{ss}^\top}{\|\mathbf{s}\|_2^2}\right)\mathbf{g}. \tag{8}$$

*Remark* 2. In particular, $T(\mathbf{m}; \pi_t)\Delta t$ represents the **optimal local transition** at the microscope. (8) is a weighted sum of the ascending direction of sparsity $\mathbf{s}$ and the descending direction of the overall energy $\mathbf{g}$, where the second term is obtained by scaling the projection of $\mathbf{g}$ onto the orthogonal space of $\mathbf{s}$. When $\mathbf{g}$ aligns with $\mathbf{s}$, the second term becomes zero and $T(\mathbf{m}; \pi_t)$ degenerates to $\mathbf{s}/\|\mathbf{s}\|^2$; otherwise, it identifies the local displacement that best aligns with $\mathbf{g}$ with a $\Delta t$-sparsity increment.

Noticeably, $T(\mathbf{m}; \pi_t)$ depends on **both the microscopic coordinate $\mathbf{m}$ and the macroscopic distribution** $\pi_t$ at sparsity level $t$. As illustrated in Figure 1, following the optimal local transition, $\pi_{t+\Delta t}$ can be obtained from $\pi_t$ by carefully adjusting each mask in support of $\pi_t$ towards the nearest optimum at the sparsity level $(t + \Delta t)$. The readily accessible $\pi_t$ provides explicit guidance for transitioning from $\pi_t$ to $\pi_{t+\Delta t}$ by refining the quality assignments of microscopic masks. This process significantly reduces the intractability of solving $\pi_{t+\Delta t}$ from scratch at a macroscopic level. As $\nu_t * \pi_t$ provides a satisfactory estimation of $\pi_{t+\Delta t}$, we are now ready to derive the distributional transition $\mathcal{T}[\cdot]$ in (2) and establish the SFPK. The proof is detailed in Appendix D.1.

**Proposition 3** (Sparsity Evolutionary Fokker-Planck-Kolmogorov Equation (informal version of Proposition 6)). *Under some regularity conditions on $L_\varepsilon(\cdot)$ and $s(\cdot)$, by taking $\Delta t \to 0$, the sequence $\{\widetilde{\pi}_{k\Delta t}\}$ constructed by $\widetilde{\pi}_{t+\Delta t} \triangleq \nu_t * \widetilde{\pi}_t$ converges weakly to the **Sparsity Evolutionary Fokker-Planck-Kolmogorov Equation (SFPK)**, denoted by $\partial_t\pi_t = -\nabla \cdot [T(\cdot; \pi_t)\pi_t]$, where $\nabla \cdot [\cdot]$ is the divergence operator and $T(\cdot; \pi_t)$ is the optimal local transition defined in Proposition 2.*

*Remark* 3. Proposition 3 provides explicit guidance on minimizing the overall energy under d$t$-sparsity increment: at sparsity level $t$, the optimal local transition induces a vector field $T(\cdot; \pi_t)\pi_t$; to minimize the overall energy while arriving $M^{\rho}_{t+dt}$, the change in density $\pi_t(\mathbf{m})$ should equal the amount of $T(\cdot; \pi_t)\pi_t$-flux crossing $\mathbf{m}$, which is equivalent to the divergence of $-T(\cdot; \pi_t)\pi_t$ at $\mathbf{m}$.

### 4.3 SFPK-GUIDED PRUNING VIA PARTICLE SIMULATION

As discussed in Section 1, to conduct neural pruning, one needs to sample performant masks from $\pi_{1-d'/d}$, the destination distribution of the SFPK. However, despite SFPK enjoys an exact form, solving $t \mapsto \pi_t$ and sampling from $\pi_{1-d'/d}$ remain challenging. Thanks to stochastic analysis theory, we show that the SFPK is associated with a McKean-Vlasov process (Veretennikov, 2006, Equation 3.1), denoted by $(\mathbf{m}_t)_{t\in[0,1]}$, satisfying $d\mathbf{m}_t = T(\mathbf{m}_t, \text{law}(\mathbf{m}_t))dt$ and $\text{law}(\mathbf{m}_t) = \pi_t$ (see Appendix D.2). Thereby, the task of sampling masks from $\pi_{1-d'/d}$ is reduced to a stochastic process simulation problem, which can be effectively accomplished by simulating a finite interacting mask particle system.

To achieve this, we propose the SFPK-pruner, which is a particle simulation-based probabilistic pruning algorithm. As outlined in Algorithm 1 (a detailed version is presented in Algorithm 2 and Algorithm 3 in Appendix A.1), the SFPK-pruner aims to simulate the evolution of $t \mapsto \pi_t$ using the evolution of the uniform distribution of $n$ interacting mask particles. At sparsity level $t$, each mask particle is updated following $d\mathbf{m}^i_t = T(\mathbf{m}^i_t; \widehat{\pi}_t)dt$, where $\widehat{\pi}_t$ denotes the uniform distribution over $\{\mathbf{m}^i_t\}^n_{i=1}$. Each gradient term in (8) is evaluated based on a mini-batch stochastic gradient. As the sparsity increases from 0 to $1 - d'/d$, the empirical distribution $\widehat{\pi}_t$ turns out to be an effective approximation of $\pi_{1-d'/d}$. Finally, we average over the empirical masks and prune the parameters associated with the smallest entries of the averaged mask. We refer the readers to Appendix A.1 for a detailed complexity analysis on SFPK-pruner.

In intuition, the SFPK-pruner can be interpreted as running an interacting ensemble of $n$ mask trajectories with increasing sparsity. While each mask is pulled by an attraction force (blue arrows in Figure 1) from the local optimum of the next sparsity level, the interaction energy $\mathcal{R}[\cdot]$ introduces repulsive forces (red arrows in Figure 1) between pairs of trajectories, which encourages the trajectories to explore diverse optima. This exploration effect is further enhanced by the gradient noise induced by the mini-batch gradient computation. In practice, our SFPK-pruner is a model-agnostic method since it does not rely on a specific formulation of the model loss with respect to the model parameters and masks. Additionally, one can implement the mask variable as either filter masks, channel masks, or node masks to conduct structured pruning with desired granularity.

## 5 EXPERIMENTS

We evaluate our SFPK-pruner for unstructured and structured pruning in one-shot and gradual pruning settings across various deep vision models. The evaluation process of each pruning method involves 3 steps: 1. Prune the well-trained target network $\boldsymbol{\theta}^*$ to a target sparsity level. 2. Retrain the pruned model for a fixed number of epochs until convergence. 3. Report the averaged prediction accuracy over the last few retraining epochs. The experiment settings are detailed in Appendix A.2.

### 5.1 EXPERIMENTS ON CIFAR-100

We conduct a comparison of the one-shot pruning performance of our SFPK-pruner against several competitive and popular baseline methods: magnitude pruning (Han et al., 2015), SNIP (Lee et al., 2019), SynFlow (Tanaka et al., 2020), GraSP (Wang et al., 2020) and PSO (Mo et al., 2023) on CIFAR-100 (Krizhevsky, 2009). We evaluate the pruning performance of each method at various sparsity levels on 3 representative models: ResNet-20 (He et al., 2016), VGG16-bn (Simonyan & Zisserman, 2015) and WRN-20 (Zagoruyko & Komodakis, 2016). For all experiments, we set $\lambda = 0.2$, $n = 10$, and $K = 100$ for SFPK-pruners.

For unstructured pruning, as shown in Figure 2, our SFPK-pruner achieves either the best or comparable performance across all the architectures and sparsity levels. Even when ranking the averaged masks of all parameters globally, our SFPK-pruner demonstrates significant performance improvements in highly sparse scenarios. This empirical evidence confirms its ability to prevent layer col-

lapse (i.e., eliminating an entire layer) in extreme compression scenarios (Tanaka et al., 2020). We further compare the channel pruning performance of the baseline methods in Figure 2. Once again, our SFPK-pruner demonstrates competitive performance across all three models and sparsity levels, indicating its capability to generate high-quality structured masks.

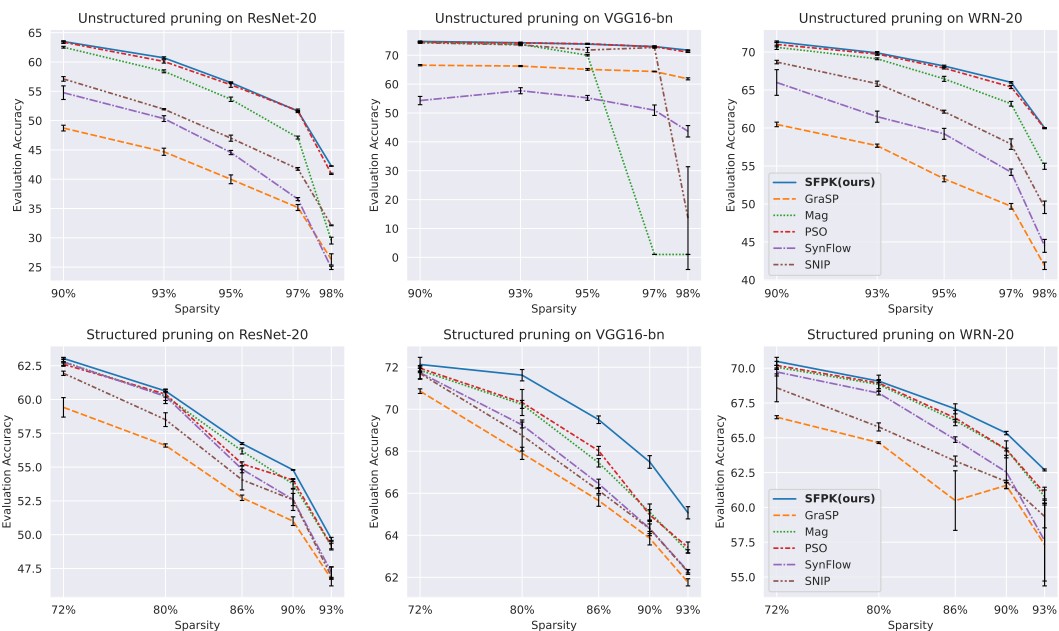

Figure 2: Top-1 accuracy (%) of one-shot pruning on CIFAR-100 over 3 random runs.

Table 1: We report the top-1 accuracy (%) of one-shot unstructured pruning **without any retraining** on ImageNet-1K. The top-1 accuracy of the unpruned models is reported in the second row. Boldface indicates the highest top-1 accuracy of each sparsity level.

| Model | | | MobileNet-V1 (72.00) | | | | | | ResNet-50 (77.01) | | | |
|---|---|---|---|---|---|---|---|---|---|---|---|---|
| Sparsity | Mag | SNIP | SynFlow | WF | PSO | **SFPK (ours)** | Mag | SNIP | SynFlow | WF | PSO | **SFPK (ours)** |
| 75% | 49.58 | 4.08 | 0.06 | 60.95 | 48.47 | **61.57** | 58.25 | 5.87 | 1.08 | 67.02 | 67.04 | **69.69** |
| 80% | 36.44 | 0.99 | 0.10 | 52.65 | 44.43 | **56.71** | 42.97 | 1.54 | 0.54 | 58.72 | 63.38 | **67.42** |
| 84% | 19.40 | 0.44 | 0.09 | 41.10 | 31.61 | **51.31** | 24.03 | 0.68 | 0.34 | 46.75 | 58.98 | **63.38** |
| 87% | 5.77 | 0.29 | 0.11 | 25.21 | 27.30 | **47.40** | 9.20 | 0.36 | 0.19 | 32.21 | 54.25 | **59.71** |
| 90% | 0.72 | 0.18 | 0.11 | 12.56 | 20.53 | **44.06** | 3.72 | 0.39 | 0.15 | 17.48 | 49.08 | **55.48** |

## 5.2 EXPERIMENTS ON IMAGENET-1K

**One-shot Pruning.** We compare the one-shot unstructured pruning effectiveness of our SFPK-pruner against two strong baselines, magnitude pruning (Han et al., 2015) and WoodFisher (denoted by WF) (Singh & Alistarh, 2020), on the ImageNet-1K benchmark (Deng et al., 2009) using two representative architectures: ResNet-50 (He et al., 2016) and MobileNet-V1 (Howard et al., 2017). For each baseline method, we directly prune the model to each density level without any retraining and then report the resulting top-1 accuracy. For SFPK-pruner, we only apply it once to generate $n$ mask trajectories terminating at 90% sparsity. We then collect intermediate checkpoints of these mask trajectories at various sparsity levels and prune the model accordingly based on the averaged mask at each intermediate sparsity level without any retraining. We ran the SFPK-pruner on all the experiments with $\lambda = 0.2$, $n = 10$, and $K = 150$. As shown in Table 1, our SFPK-pruner consistently outperforms other baselines at all sparsity levels. This validates the effectiveness of our proposed SFPK in approximating the sparsity evolution of the optimal mask distribution. Notably, the computation cost of running the SFPK-pruner with a batch size of 256 for $n \times K = 1500$ steps is equivalent to that of only 0.3 epoch of retraining on ImageNet. Thus, the particle simulation-based

SFPK-pruner enables us to sample performant masks with a modest number of particles, avoiding excessive overhead.

We also evaluate the one-shot structured channel pruning performance of our SFPK-pruner on the transformer-based DeiT-T architecture (Touvron et al., 2021) against the state-of-the-art pruned transformer, including SCOP (Tang et al., 2020), HVT (Pan et al., 2021), UVC (Yu et al., 2022b), WDPruning (Yu et al., 2022a), and X-Pruner (Yu & Xiang, 2023). We apply the SFPK-pruner to prune the model to 50% sparsity with $\lambda = 0.2$, $n = 10$, $K = 1000$. Then, we retrain the pruned model for 100 epochs following (Yu & Xiang, 2023). As shown in Table 2a, our SFPK-pruner achieves competitive top-1 accuracy compared to baseline methods within a comparable FLOP budget. This showcases the efficacy of our SFPK-pruner in structured pruning for transformers.

Table 2: For each experiment, we report the top-1 accuracy (%) of the pruned model. In Table 2b and Table 2c, we further report the relative performance drop (%) compared to the dense model.

(a) One-shot structured pruning on DeiT-T.

| ImageNet-1K | One-shot Structured Pruning | | |
|---|---|---|---|
| DeiT-T | Acc@1 | Acc@5 | FLOPs (%) |
| Unpruned | 72.2 | 91.1 | 100.0 |
| SCOP | 68.9 | 89.0 | 61.5 |
| HVT | 69.7 | 89.4 | 53.8 |
| PSO | 69.9 | 89.4 | 52.1 |
| WDPruning | 70.3 | 89.8 | 53.8 |
| UVC | 70.6 | - | 39.1 |
| X-Pruner | 71.1 | 90.1 | 49.2 |
| **SFPK (ours)** | **71.6** | **90.3** | 51.7 |

(b) Gradual pruning on MobileNet-V1.

| ImageNet-1K | Gradual Unstructured Pruning | | | |
|---|---|---|---|---|
| MobileNet-V1 | Dense | Pruned | Rel. Drop | Sparsity |
| Incremental | 70.60 | 67.60 | -4.25 | 74.11 |
| STR | 72.00 | 68.35 | -5.07 | 75.28 |
| Mag | 72.00 | 69.90 | -2.92 | 75.28 |
| PSO | 72.00 | 69.16 | -3.95 | 75.28 |
| WF | 72.00 | **70.09** | **-2.65** | 75.28 |
| **SFPK (ours)** | 72.00 | 69.79 | -3.07 | 75.00 |
| Incremental | 70.60 | 61.80 | -12.46 | 89.03 |
| STR | 72.00 | 62.10 | -13.75 | 89.01 |
| Mag | 72.00 | 63.02 | -12.47 | 89.00 |
| PSO | 72.00 | 63.64 | -11.61 | 90.00 |
| WF | 72.00 | 63.87 | -11.29 | 89.00 |
| **SFPK (ours)** | 72.00 | **64.10** | **-10.97** | 90.00 |

(c) Gradual pruning on ResNet-50.

| ImageNet-1K | Gradual Unstructured Pruning | | | |
|---|---|---|---|---|
| ResNet-50 | Dense | Pruned | Rel. Drop | Sparsity |
| DSR | 74.90 | 71.60 | -4.41 | 80.00 |
| Incremental | 75.95 | 74.25 | -2.24 | 73.50 |
| DPF | 75.95 | 75.13 | -1.08 | 79.90 |
| GMP + LS | 76.69 | 75.58 | -1.45 | 79.90 |
| VD | 76.69 | 75.28 | -1.84 | 80.00 |
| RIGL + ERK | 76.80 | 75.10 | -2.21 | 80.00 |
| SNFS + LS | 77.00 | 74.90 | -2.73 | 80.00 |
| STR | 77.01 | 76.19 | -1.06 | 79.55 |
| PSO | 77.01 | 75.84 | -1.52 | 80.00 |
| DNW | 77.50 | 76.20 | -1.68 | 80.00 |
| WF | 77.01 | **76.76** | **-0.32** | 80.00 |
| CrAM | 77.3 | 75.80 | -1.94 | 90.00 |
| **SFPK (ours)** | 77.01 | 76.18 | -1.26 | 80.00 |
| Mag | 77.01 | 75.15 | -2.42 | 90.00 |
| GMP + LS | 76.69 | 73.91 | -3.62 | 90.00 |
| VD | 76.69 | 73.84 | -3.72 | 90.27 |
| RIGL + ERK | 76.80 | 73.00 | -4.95 | 90.00 |
| SNFS + LS | 77.00 | 72.90 | -5.32 | 90.00 |
| STR | 77.01 | 74.31 | -3.51 | 90.23 |
| PSO | 77.01 | 74.63 | -3.09 | 90.00 |
| DNW | 77.50 | 74.00 | -4.52 | 90.00 |
| WF | 77.01 | 75.21 | -2.34 | 90.00 |
| CrAM | 77.3 | 74.70 | -4.66 | 90.00 |
| **SFPK (ours)** | 77.01 | **75.24** | **-2.30** | 90.00 |

**Gradual Pruning.** To further demonstrate the effectiveness of our SFPK-pruner, we evaluate its gradual pruning performance on the pre-trained ResNet-50 and MobileNet-V1 models provided in (Kusupati et al., 2020). We apply the SFPK-pruner to gradually prune the model to 90% sparsity within 10 prune-and-retrain loops. Specifically, the target sparsity and the number of retraining epochs of each pruning shot increase exponentially, while the total retraining epochs are kept at 100 to ensure a fair comparison with other baseline methods including Mag (Han et al., 2015), Incremental (Zhu & Gupta, 2018b), DPF (Lin et al., 2020), GMP+LS (Gale et al., 2019), VD (Molchanov et al., 2017b), RIGL+ERK (Evci et al., 2020), SNFS+LS (Dettmers & Zettlemoyer, 2020), STR (Kusupati et al., 2020), DNW (Wortsman et al., 2019), WF (Singh & Alistarh, 2020), PSO (Mo et al., 2023), and CrAM (Peste et al., 2023). We refer the reader to Appendix C for additional experiments under different training budgets and pretrained checkpoints. We fix $\lambda = 0.2$, $n = 10$, $K = 150$ for each pruning shot. In Table 2b and Table 2c, our SFPK-pruner demonstrates decent performance compared to various competitive pruning baselines. Notably, the intermediate sparsity checkpoints of our SFPK-pruner undergo fewer retraining epochs (as the total retrain epochs are fixed to 100) than other baseline methods. This validates the effectiveness of our SFPK in guiding the gradual sparsification of the model.

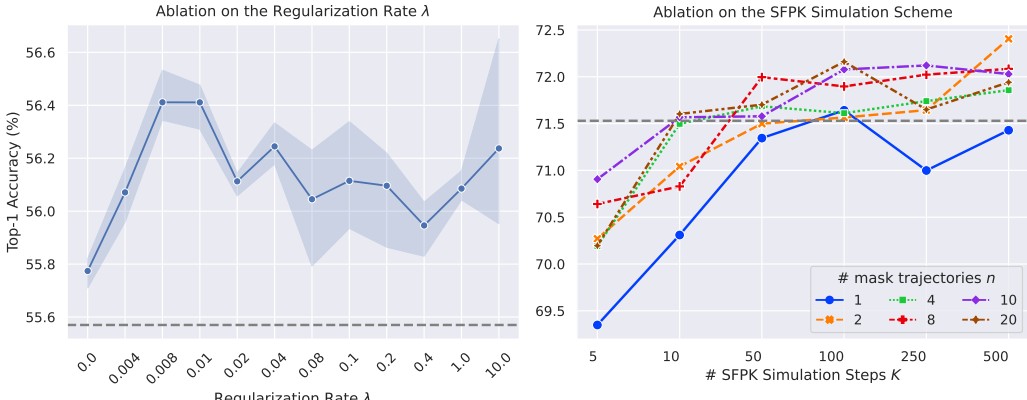

Figure 3: Top-1 accuracy (%) for each experiment. The grey horizontal dashed line indicates the performance of iterative magnitude pruner (Zimmer et al., 2022) under the same settings.

## 5.3 ABLATION STUDIES

**Ablation on the Regularization Rate $\lambda$.** As shown in Definition 1, the convexification technique inevitably introduces a small gap proportional to $\lambda$ between the original pruning objective and the convexified one. To test the sensitivity of SFPK-pruner's performance to $\lambda$, we implement the one-shot SFPK-pruner with different $\lambda$ values to compress a pretrained ResNet-20 to $95\%$ sparsity on CIFAR-100. Figure 3 (top) shows that our SFPK-pruner is robust against various $\lambda$ values and performs well even with small $\lambda$. We hypothesize that this is because we use mini-batch stochastic gradients to approximate the gradient terms in (8), implicitly introducing gradient noise that acts as a repulsive random force, serving as a surrogate for the convex regularizer.

**Ablation on the Mask Particle Simulation Scheme.** According to Algorithm 1, our SFPK-pruner relies on numerically simulating the stochastic process $d\mathbf{m}_t = T(\mathbf{m}, \text{law}(\mathbf{m}_t))dt$ using $n$ interacting mask particles. The discretization of the sparsity horizon (controlled by simulation steps $K$) and the spatial horizon (controlled by $n$) are crucial to SFPK's performance. We apply a one-shot SFPK-pruner with various $(K, n)$ values to compress a pretrained VGG16-bn to $98\%$ sparsity on CIFAR-100. Figure 3 (bottom) shows that larger $(K, n)$ leads to finer simulation resolution and higher performance. In practice, our SFPK-pruner effectively approximates the optimal mask distribution's sparsity evolution with $n \geqslant 4$ and $K \geqslant 50$.

## 6 CONCLUSIONS

In this paper, we analyze the dynamics of the population of optimal neural pruning masks as sparsity increases. We propose the Sparsity Evolutionary Fokker-Planck-Kolmogorov Equation (SFPK) as an approximation for the evolution of mask distribution. Additionally, we introduce a probabilistic pruning method called SFPK-pruner, which uses an interacting mask particle system to sample performant masks at desired sparsity levels. The acceleration and scaling of SFPK-pruner for larger models will be addressed in future work.

## ACKNOWLEDGEMENT

The research work described in this paper was conducted in the JC STEM Lab of Machine Learning and Symbolic Reasoning funded by The Hong Kong Jockey Club Charities Trust. Sinno J. Pan also thanks the support of the Microsoft Research Asia collaborative research grant.

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

# A    IMPLEMENTATION DETAILS

## A.1    IMPLEMENTATION DETAILS OF SFPK-PRUNER

**Implementation of SFPK-pruner.** A detailed implementation of the SFPK-pruner (Algorithm 1) is shown in Algorithm 2. A PyTorch implementation is attached in Algorithm 3.

Notably, the mini-batch gradients involved in Algorithm 2 are crucial for mask particles' simulation. In theory, the SFPK dynamic should be initialized at $\pi_0^*$, the optimal mask distribution within $M_0^\rho$. Since $\rho > 0$, $\pi_0^*$ is close to, but not identical to, the delta distribution centered on the fully dense mask. In practice, we approximate $\pi_0^*$ by running SFPK with stochastic mini-batch gradient from the delta distribution at the fully dense model. During the SFPK particle simulation process, the stochasticity of the mini-batch gradients enhances robustness and stability in sampling, while also encouraging the exploration of diverse mask particles.

**Complexity analysis.** As shown in Algorithm 3, the extra computation is `SFPK_prune()`. It consists of $n \times K$ inner SFPK steps. Each SFPK step includes one forward-backward pass on a mini-batch (to calculate `g`) and the computation of sparsity gradient `s`, transition `trans`, and mask update. Let $d$ be the number of parameters of the model and $s(m) = 1 - \|m\|_2^2/d$ be the soft sparsity function, `s` can be computed in closed-form in $O(d)$. Computing `trans` evolves only inner-products with $O(d)$ complexity, and the mask update also has a complexity of $O(d)$. The forward-backward pass dominates the cost of one SFPK step. The cost of one `SFPK_prune()` call equals the cost of "training the model for $n \times K$ mini-batches" plus a marginal $O(nKd)$ inner-product computation.

In our ImageNet experiments, we set $n = 150$, $K = 10$, and batch-size $= 256$. Thus, each `SFPK_prune()` call is as cheap as $1,500$ training steps, which equals $0.3$ epochs (1 epoch of ImageNet has $1,281,167/256 \approx 5000$ batches). As we perform 10 prune-and-retrain iterations, the total SFPK overhead is as inexpensive as merely $0.3 \times 10 = 3$ additional training epochs. This is marginal when compared to the retraining process(100-epoch) of the baseline. In Table 12, Table 14 and Table 16, we compare the training and inference FLOPs of SFPK with other baselines.

**Constraints of SPFK-pruner.** Our SFPK works effectively when the target model is "rational," meaning pruning the model usually increases the loss (i.e. harms the performance) rather than decreases it (i.e. improves the performance). For instance, an adversarially trained model (e.g., trained to maximize task loss) is considered "irrational", as nullifying some parameters may reduce the task loss. In this case, applying SFPK to this model can still progressively improve its accuracy as sparsity increases, because the SFPK update (8) always pushes the particles towards smaller loss. However, the SFPK trajectory may NOT converge to the true optimal mask distribution dynamic $t \mapsto \pi_t$, which may lead to suboptimal results.

In intuition, this is because the dense mask $m = (1, ..., 1)$ in this scenario is no longer the optimal mask across all sparsity levels, and it fails to guide the distributional transition of the optimal mask. Therefore, the SFPK trajectory may diverge from $t \mapsto \pi_t$ as it gets lost from the beginning $t = 0$. In theory, SFPK might fail to approximate $t \mapsto \pi_t$ as the assumptions (A1) and (A2) in Proposition 4 are violated. Hence, the convergence theorem no longer holds. We recommend applying SFPK to sufficiently trained "rational" models (e.g. trained to convergence), where the dense mask is likely to be close to the optimal mask across all sparsity levels.

## A.2    IMPLEMENTATION OF PRUNING EXPERIMENTS

**Unstructured and Structured Pruning.** The only difference between the unstructured and structured pruning is the granularity of pruning. In unstructured pruning, each parameter is prunable. In contrast, structured pruning targets substructures of the neural network, such as all the weights associated with a kernel, all the weights associated with a channel, or all the weights associated with an attention head. In this paper, we only consider structured channel pruning, where the weights bonded to a particular channel are governed by a single pruning mask. For pruning methods initially devised for unstructured pruning, we adapt them to the structured pruning scenario by calculating the importance score of a channel based on the sum of the importance scores of all its associated weights.

**One-shot and Gradual Pruning.** We consider a pruning shot that involves three steps: 1. Prune the target model $\boldsymbol{\theta}^*$ to the target sparsity level $t$. 2. Retrain the pruned model for $N$ epochs to regain performance. 3. Evaluate and report the top-1 predictive accuracy of the retrained model. Given a target sparsity level $t$, the one-shot pruning procedure involves a single pruning step. In contrast, the gradual pruning procedure consists of multiple pruning steps, with each step compressing the model toward a pre-specified intermediate sparsity level until the target sparsity is achieved. These intermediate sparsity levels represent the sparsity schedule. We use an exponential sparsity schedule across all the gradual pruning experiments: for a $T$-shot gradual pruning experiment, the $i$-th intermediate sparsity level is $t^{i/K}$. Thus, the configuration of a one-shot or gradual pruning experiment is characterized by the pruning method and the retraining scheme, which includes the epoch amount, learning rate schedule, batch size, and optimizer. The configurations of all pruning experiments are elaborated in Table 3 and Table 4.

## B  FULL EXPERIMENT RESULTS

In this section, we present the complete experimental results from Section 5, including error bars representing standard deviations, along with results from additional settings. All reported results are based on 3 independent trials, following the experimental setup outlined in Appendix A.2. The standard deviations of the gradual pruning results of our SFPK-pruner are reported in Table 5.

Table 5: For each experiment, we report the top-1 accuracy (%) of the pruned model. In Table 5b and Table 5c, we further report the relative performance drop (%) compared to the dense model.

(a) One-shot structured pruning on DeiT-T.

| ImageNet-1K | One-shot Structured Pruning | | |
|---|---|---|---|
| DeiT-T | Acc@1 | Acc@5 | FLOPs (%) |
| Unpruned | 72.2 | 91.1 | 100.0 |
| SCOP | 68.9 | 89.0 | 61.5 |
| HVT | 69.7 | 89.4 | 53.8 |
| PSO | 69.9 | 89.4 | 52.1 |
| WDPruning | 70.3 | 89.8 | 53.8 |
| UVC | 70.6 | - | 39.1 |
| X-Pruner | 71.1 | 90.1 | 49.2 |
| **SFPK (ours)** | **71.6 (0.2)** | **90.3 (0.1)** | 51.7 |

(b) Gradual pruning on MobileNet-V1.

| ImageNet-1K | Gradual Unstructured Pruning | | | |
|---|---|---|---|---|
| MobileNet-V1 | Dense | Pruned | Rel. Drop | Sparsity |
| Incremental | 70.60 | 67.60 | -4.25 | 74.11 |
| STR | 72.00 | 68.35 | -5.07 | 75.28 |
| Mag | 72.00 | 69.90 | -2.92 | 75.28 |
| PSO | 72.00 | 69.16 | -3.95 | 75.28 |
| WF | 72.00 | **70.09** | **-2.65** | 75.28 |
| **SFPK (ours)** | 72.00 | 69.79 (0.24) | -3.07 (0.29) | 75.00 |
| Incremental | 70.60 | 61.80 | -12.46 | 89.03 |
| STR | 72.00 | 62.10 | -13.75 | 89.01 |
| Mag | 72.00 | 63.02 | -12.47 | 89.00 |
| PSO | 72.00 | 63.64 | -11.61 | 90.00 |
| WF | 72.00 | 63.87 | -11.29 | 89.00 |
| **SFPK (ours)** | 72.00 | **64.10 (0.17)** | **-10.97 (0.19)** | 90.00 |

(c) Gradual pruning on ResNet-50.

| ImageNet-1K | Gradual Unstructured Pruning | | | |
|---|---|---|---|---|
| ResNet-50 | Dense | Pruned | Rel. Drop | Sparsity |
| DSR | 74.90 | 71.60 | -4.41 | 80.00 |
| Incremental | 75.95 | 74.25 | -2.24 | 73.50 |
| DPF | 75.95 | 75.13 | -1.08 | 79.90 |
| GMP + LS | 76.69 | 75.58 | -1.45 | 79.90 |
| VD | 76.69 | 75.28 | -1.84 | 80.00 |
| RIGL + ERK | 76.80 | 75.10 | -2.21 | 80.00 |
| SNFS + LS | 77.00 | 74.90 | -2.73 | 80.00 |
| STR | 77.01 | 76.19 | -1.06 | 79.55 |
| PSO | 77.01 | 75.84 | -1.52 | 80.00 |
| DNW | 77.50 | 76.20 | -1.68 | 80.00 |
| WF | 77.01 | **76.76** | **-0.32** | 80.00 |
| CrAM | 77.3 | 75.80 | -1.94 | 90.00 |
| **SFPK (ours)** | 77.01 | 76.18 (0.13) | -1.26 (0.14) | 80.00 |
| Mag | 77.01 | 75.15 | -2.42 | 90.00 |
| GMP + LS | 76.69 | 73.91 | -3.62 | 90.00 |
| VD | 76.69 | 73.84 | -3.72 | 90.27 |
| RIGL + ERK | 76.80 | 73.00 | -4.95 | 90.00 |
| SNFS + LS | 77.00 | 72.90 | -5.32 | 90.00 |
| STR | 77.01 | 74.31 | -3.51 | 90.23 |
| PSO | 77.01 | 74.63 | -3.09 | 90.00 |
| DNW | 77.50 | 74.00 | -4.52 | 90.00 |
| WF | 77.01 | 75.21 | -2.34 | 90.00 |
| CrAM | 77.3 | 74.70 | -4.66 | 90.00 |
| **SFPK (ours)** | 77.01 | **75.24 (0.09)** | **-2.30 (0.09)** | 90.00 |

**One-shot pruning on ImageNet-1K.** Following the experimental setup in Section 5.2, we perform one-shot unstructured pruning experiments on the ImageNet-1K (Deng et al., 2009) dataset using ResNet-50 (He et al., 2016), MobileNet-V1 (Howard et al., 2017), and DeiT-T (Touvron et al., 2021) architectures. We compare our SFPK-pruner against several strong baselines, including magnitude pruning (Han et al., 2015), SNIP (Lee et al., 2019), (Tanaka et al., 2020), WoodFisher (Singh & Alistarh, 2020), and PSO (Mo et al., 2023). As shown in Table 6, our SFPK-pruner exhibits high performance across various architectures. Our experiment results validate that our SFPK effectively approximates the sparsity evolution of the optimal mask distribution, enabling us to sample high-quality masks within acceptable overhead.

---

**Algorithm 2** SFPK-guided Pruning via Particle Simulation (SFPK-pruner)

---

1: **Input:** target budget $d'$, dataset $S$, batch size $b$, target model parameterized by $f(\cdot; \boldsymbol{\theta}^*)$, model loss $L(\cdot, \cdot)$, soft sparsity function $s(\cdot)$, RBF kernel $\kappa(\cdot, \cdot)$, mask polarizer $P_\epsilon(\cdot)$, localization radius $r_t$, particle number $n$, simulation steps $K$, regularization rate $\lambda$.
2: **Output:** a performant binary mask $\widehat{\mathbf{m}} \in \{0, 1\}^d$ with $\|\widehat{\mathbf{m}}\|_0 = d'$.
3: $t \leftarrow 0, \Delta t \leftarrow (1 - d'/d)/K$.
4: Initialize $\mathbf{m}_t^i \leftarrow \mathbf{1}$, $i = 1, ..., n$.
5: **for** $k = 1$ **to** $K$ **do**
6:     **for** $i = 1$ **to** $n$ **do**
7:         $\mathbf{s} \leftarrow \nabla s(\mathbf{m}_t^i)$
8:         $(\mathbf{x}, \mathbf{y}) \leftarrow \text{Mini\_Batch}(S; b)$
9:         $\widehat{\mathbf{g}} \leftarrow -\partial_{\mathbf{m}} L(f(\mathbf{x}; P_\varepsilon(\mathbf{m}_t^i) \odot \boldsymbol{\theta}^*), \mathbf{y}) - \frac{\lambda}{n} \sum_{j=1}^n \partial_1 \kappa(\mathbf{m}_t^i, \mathbf{m}_t^j)$
10:         $\widehat{T}(\mathbf{m}_t^i) \leftarrow \mathbf{s}/\|\mathbf{s}\|_2^2 + \left( (r_t^2 \|\mathbf{s}\|_2^2 - 1) / (\|\mathbf{s}\|_2^2 \|\widehat{\mathbf{g}}\|_2^2 - (\mathbf{s}^\top \widehat{\mathbf{g}})^2) \right)^{\frac{1}{2}} \cdot (\widehat{\mathbf{g}} - (\widehat{\mathbf{g}}^\top \mathbf{s})/\|\mathbf{s}\|^2 \cdot \mathbf{s}) \widehat{\mathbf{g}}$
11:         $\mathbf{m}_{t+\Delta t}^i \leftarrow \mathbf{m}_t^i + \widehat{T}(\mathbf{m}_t^i)\Delta t$
12:     **end for**
13:     $t \leftarrow t + \Delta t$
14: **end for**
15: $\widehat{\mathbf{m}} \leftarrow \text{top}_{d'} \left( \sum_{i=1}^n \mathbf{m}_t^i \right)$
16: **return** $\widehat{\mathbf{m}}$

---

**Algorithm 3** SFPK-pruner in PyTorch

---

```python
def main(model, t, m, dataloader, **args):
    # model: masked dense model
    # t: target sparsity
    # m: pruning steps
    # args: SFPK pruning args
    for i in range(m):
        t_tmp = i_th_sparsity(t, i) # gradual pruning schedule
        # model = baseline_prune(model, t_tmp, dataloader, **args)
        model = SFPK_prune(model, t_tmp, dataloader, **args)
        model = retrain(model)

    return model

def SFPK_prune(model, t, dataloader, n, K, r):
    # model: masked model
    # t: target sparsity
    # n: particle number
    # K: simulation steps
    # r: local radius
    eps = 1e-18
    masks = init_masks(model, n)
    dt = (t - sparsity(model)) / K
    for i in range(K):
        for mask in masks:
            s = calc_s(mask) # calc grad of soft sparsity (1 - norm(mask)) explicitly
            x, y = next(dataloader)
            reg = calc_kappa(mask, masks) # calc interaction penalty
            loss = loss_fn(model, mask, x, y) + reg
            g = - grad(loss, mask) # calc grad w.r.t mask

            # calc transition
            gap = (norm(s) * norm(g)) ** 2 - inner_prod(s, g) ** 2
            if gap > eps:
                term1 = s / norm(s) ** 2
                scale = (((r * norm(s)) ** 2 - 1) / gap) ** 0.5
                term2 = scale * (g - inner_prod(s, g) * term1)
                trans = term1 + term2
            else:
                # term2 equals 0 if g is parallel to s
                trans = s / norm(s) ** 2

            mask += trans * dt # SFPK update

    return prune(model, mean(masks))
```

---

Table 3: Experiment settings of one-shot pruning on CIFAR-100. Here, "lr" denotes "learning rate", and "Cos" denotes the cosine annealing learning rate schedule, with an initial learning rate of 1e-4 and 5 warm-up epochs. We follow the implementation of mask polarizer as (Mo et al., 2023, Algorithm 3). Each run takes around 20 to 30 minutes on a NVIDIA 40G A100 GPU, with each SFPK update step (line 7 to line 11 of Algorithm 2) taking approximately 0.2 seconds.

| CIFAR-100 | | Structured | | | Unstructured | | |
|---|---|---|---|---|---|---|---|
| | Model | ResNet-20 | VGG16-bn | WRN-20 | ResNet-20 | VGG16-bn | WRN-20 |
| | Top-1 accuracy | 70.38 | 75.68 | 75.22 | 70.38 | 75.68 | 75.22 |
| Retrain setting | Batch size | 64 | 64 | 64 | 64 | 64 | 64 |
| | Epochs | 100 | 100 | 100 | 100 | 100 | 100 |
| | lr | 0.01 | 0.01 | 0.01 | 0.01 | 0.01 | 0.01 |
| | lr schedule | Cos | Cos | Cos | Cos | Cos | Cos |
| | Optimizer | SGD | SGD | SGD | SGD | SGD | SGD |
| | Momentum | 0.875 | 0.875 | 0.875 | 0.875 | 0.875 | 0.875 |
| | Weight decay | 0 | 0 | 0 | 0 | 0 | 0 |
| SFPK setting | Mini-batch size | 512 | 512 | 512 | 512 | 512 | 512 |
| | Simulation steps $K$ | 100 | 100 | 100 | 100 | 100 | 100 |
| | # mask particles $n$ | 10 | 10 | 10 | 10 | 10 | 10 |
| | Regularization rate $\lambda$ | 0.2 | 0.2 | 0.2 | 0.2 | 0.2 | 0.2 |
| | Localization radius $r_t$ | 1.1 | 1.1 | 1.01 | 1.1 | 1.1 | 2 |
| | Mask polarizer $P_\varepsilon$ | One-hot | One-hot | One-hot | One-hot | One-hot | One-hot |

Table 4: Experiment settings of pruning experiments on ImageNet-1K. Here, "lr" denotes "learning rate", and "Cos" denotes the cosine annealing learning rate schedule **without any warm-up epoch**. We follow the implementation of mask polarizer as (Mo et al., 2023, Algorithm 3). The one-shot pruning without retraining experiments shown in Table 1 follow the same SFPK settings as described above. On an NVIDIA 40G A100 GPU, each run on MobileNet-1K takes about 1.5 days, each run on ResNet-50 takes about 2 days, and each run on DeiT-T takes about 4 days. Each SFPK update step (lines 7 to 11 of Algorithm 2) takes approximately 0.4 seconds.

| Pruning on ImageNet-1K | | | | |
|---|---|---|---|---|
| | Model | ResNet-50 | MobileNet-V1 | DeiT-T |
| | Top-1 accuracy | 77.01 | 72.00 | 72.20 |
| | Granularity | Unstructured | Unstructured | Structured |
| | # pruning shots | 10 | 10 | 1 |
| Retrain setting | Batch size | 256 | 256 | 512 |
| | Epochs | 100 | 100 | 100 |
| | lr | 1e-2 | 1e-2 | 1e-4 |
| | lr schedule | Cos | Cos | Cos |
| | Optimizer | SGD | SGD | AdamW |
| | Momentum | 0.875 | 0.875 | 0.875 |
| | Weight decay | 2e-5 | 2e-5 | 5e-2 |
| | Label smoothing | 0.1 | 0.1 | 0.1 |
| SFPK setting | Mini-batch size | 256 | 256 | 256 |
| | Simulation steps $K$ | 150 | 150 | 1000 |
| | # mask particles $n$ | 10 | 10 | 10 |
| | Regularization rate $\lambda$ | 0.2 | 0.2 | 0.2 |
| | Localization radius $r_t$ | 3 | 4 | 8 |
| | Mask polarizer $P_\varepsilon$ | One-hot | One-hot | One-hot |

| ResNet-50 (77.01) | | | | | |
|---|---|---|---|---|---|
| Sparsity | Mag | SNIP | SynFlow | WF | PSO | **SFPK (ours)** |
| 75% | 58.25 | 5.87 | 1.08 | 67.02 | 67.04 | **69.69** |
| 80% | 42.97 | 1.54 | 0.54 | 58.72 | 63.38 | **67.42** |
| 84% | 24.03 | 0.68 | 0.34 | 46.75 | 58.98 | **63.38** |
| 87% | 9.20 | 0.36 | 0.19 | 32.21 | 54.25 | **59.71** |
| 90% | 3.71 | 0.39 | 0.15 | 17.48 | 49.08 | **55.48** |
| MobileNet-V1 (72.00) | | | | | |
| Sparsity | Mag | SNIP | SynFlow | WF | PSO | **SFPK (ours)** |
| 75% | 49.58 | 4.08 | 0.06 | 60.95 | 48.47 | **61.57** |
| 80% | 36.44 | 0.99 | 0.10 | 52.65 | 44.43 | **56.71** |
| 84% | 19.40 | 0.44 | 0.09 | 41.10 | 31.61 | **51.31** |
| 87% | 5.77 | 0.29 | 0.11 | 25.21 | 27.30 | **47.40** |
| 90% | 0.72 | 0.18 | 0.11 | 12.56 | 20.53 | **44.06** |
| Deit-T (72.20) | | | | | |
| Sparsity | Mag | SNIP | SynFlow | PSO | **SFPK (ours)** |
| 20% | 68.14 | 61.53 | 1.21 | 65.91 | **69.94** |
| 37% | 39.74 | 36.50 | 0.92 | 59.05 | **64.22** |
| 50% | 6.51 | 10.54 | 0.99 | 46.02 | **55.97** |
| 60% | 1.47 | 1.99 | 0.31 | 31.82 | **45.37** |
| 75% | 0.16 | 0.25 | 0.13 | 12.60 | **32.70** |

Table 6: We report the top-1 accuracy (%) of one-shot unstructured pruning **without any retraining** of ResNet-50 (top), MobileNet-V1 (middle), and DeiT-T (bottom) on ImageNet-1K. Boldface indicates the highest top-1 accuracy of each sparsity level.

**Comparisons on runtime complexity.** In Table 7, we report the actual and theoretical runtime complexities of WF (Singh & Alistarh, 2020), PSO (Mo et al., 2023), and our SFPK-pruner for the one-shot pruning experiment on ImageNet-1K with ResNet-50 and MobileNet-V1 at $80\%$ sparsity. Our results show that, in practice, SFPK outperforms WF while maintaining a comparable computational cost. As expected, the computational cost of SFPK is approximately 10 times that of PSO, as both methods use 700 discretization steps, while SFPK additionally simulates 10 particles.

We also compare the theoretical complexity of our SFPK-pruner with that of PSO and WF. Let $m$ denote the data batch size, $d$ the model size, $K$ the discretization step for both PSO and SFPK-pruner, and $n$ the number of particles in the SFPK-pruner. The theoretical runtime complexity of the SFPK-pruner is $O(nK(d+m))$, the theoretical runtime complexity of PSO is $O(K(d+m))$, and the theoretical runtime complexity of WF is $O(md^2)$. As we will show, for large models where $d$ is extremely large, WF becomes significantly expensive due to the need to compute the Hessian matrix. As expected, the computational cost of SFPK is approximately $n$ times that of PSO, as both methods use $K$ discretization steps, while SFPK additionally simulates $n$ particles.

| 95% sparsity | ResNet-50 on ImageNet-1K | | | MobileNet-V1 on ImageNet-1K | | |
|---|---|---|---|---|---|---|
| Method | WF | PSO | SFPK (ours) | WF | PSO | SFPK (ours) |
| Runtime (s) | 9629.23 (64.35) | 778.41 (9.06) | 7544.41 (53.28) | 3023.11 (122.41) | 368.41 (9.06) | 3066.60 (22.79) |

Table 7: Comparison of the runtime complexity. We followed the recommended configuration of WF (Singh & Alistarh, 2020) and configuration in Table 4. For each experiment, we present the average execution time over 3 random runs on an NVIDIA 40G A100 GPU, with standard deviations.

## C  ADDITIONAL EXPERIMENTS

**Visualizing the layer-wise sparsity of SFPK-pruner.** Our SFPK-pruner achieves distinct layer-wise sparsity ratios compared to ERK and other baselines (e.g., WF (Singh & Alistarh, 2020), STR (Kusupati et al., 2020), SNFS (Dettmers & Zettlemoyer, 2020)), effectively avoiding layer collapse. As shown in Figure 4, the layer-wise density (i.e. 1 - sparsity) of SFPK on ResNet-50 ImageNet

concentrates on the earlier layers, without using heuristics like "freezing the first Conv / last FC layer". Unlike ERK, which assigns higher sparsity to deeper layers, SFPK shows higher sparsity in mid-to-deep layers (IDs 30 to 45) with a smoother overall pattern. SFPK-pruner's sparsity also differs from WF's in early layers, showing that the sparsity pattern of early layers is crucial for pruning performance.

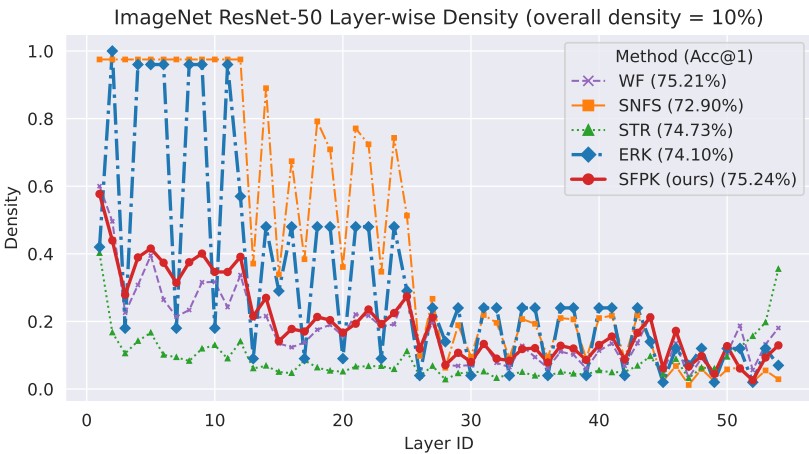

Figure 4: Visualizing the layer-wise density patterns.

**Ablation study on the rationality of $\boldsymbol{\theta}^*$.** To study how the rationality of $\boldsymbol{\theta}^*$ affects the effectiveness of SFPK-pruner, we conduct an ablation study on the initialization of SFPK-pruner on WRN-32x4 (Zagoruyko & Komodakis, 2016) CIFAR-100 (Krizhevsky, 2009). Specifically, we first pretrain a dense WRN-32x4 from scratch for 300 epochs to convergence, and we save the intermediate model checkpoints $\boldsymbol{\theta}_t^*$ at training epoch $t \in \{0, 50, 100, 150, 200, 300\}$. Then, we prune each $\boldsymbol{\theta}_t^*$ to 90% sparsity in a one-shot unstructured manner and retrain it for 100 epochs. We report the training accuracy of each setting based on 3 trials.

As shown in Table 8, the performance of SFPK increases as the pretraining epochs increase. Specifically, the performance of SFPK-pruner retrains at a decent level even when the model is not pretrained to convergence. This empirical evidence suggests the robustness of SFPK-pruner to the rationality of the model weight.

| Pretrained Epoch | 0 | 100 | 150 | 200 | 250 | 300 |
|---|---|---|---|---|---|---|
| Dense Acc. | 1.02 (0.01) | 51.05 (0.12) | 59.15 (0.20) | 66.52 (0.08) | 72.34 (0.12) | 76.01 (0.16) |
| Pruned Acc. | 1.03 (0.04) | 50.42 (0.14) | 54.07 (0.21) | 60.84 (0.11) | 62.7 (0.11) | 65.66 (0.14) |
| Tuned Acc. | 63.27 (0.52) | 68.05 (0.33) | 72.2 (0.17) | 72.42 (0.14) | 72.63 (0.09) | 73.24 (0.10) |

Table 8: Ablation study on the rationality of $\boldsymbol{\theta}^*$. We report the top-1 accuracy (%) of unstructured one-shot pruning on WRN-32x4 CIFAR-100. For each pruned model, we present the average top-1 accuracy over 3 random runs, with standard deviations in parentheses.

**Pruning at initialization with SFPK.** We compare our SFPK against SNIP (Lee et al., 2019), SynFlow (Tanaka et al., 2020), GraSP (Wang et al., 2020), and Mag (Han et al., 2015) under the pruning-at-initialization setting. Specifically, we randomly initialize a WRN-20 (Zagoruyko & Komodakis, 2016), prune it to 95% sparsity, and retrain it for 100 epochs to convergence. The average test accuracy among of final 5 epochs is reported based on 3 trials.

As shown in Table 9, our SFPK illustrates either comparable or better performance than other baselines. However, the performance of SFPK can be improved when a well-trained checkpoint is provided. This evidence helps clarify the significance of the 'rational' constraint.

| Sparsity = 95% | Dense | Mag | SNIP | GraSP | SynFlow | SFPK |
|---|---|---|---|---|---|---|
| PAI Acc. | - | 52.65 (0.21) | 54.19 (0.09) | 52.38 (0.12) | 54.97 (0.17) | **55.62 (0.11)** |
| PTP Acc. | 75.26 | 66.47 (0.33) | 62.15 (0.19) | 53.31 (0.39) | 59.23 (0.72) | **68.16 (0.11)** |

Table 9: We report the top-1 accuracy (%) of unstructured one-shot pruning at initialization (PAI) and post-training pruning (PTP) results with WRN-20 CIFAR-100. For each pruned model, we present the average top-1 accuracy over 3 random runs, with standard deviations in parentheses.

**Pruning ResNet-50 to** $95\%$ **sparsity on ImageNet-1K.** Following the setup in Table 4, we prune ResNet-50 (He et al., 2016) to the $95\%$ sparsity level on ImageNet-1K (Deng et al., 2009) using SFPK with the same hyperparameters. Due to the limited computation resource, the reported accuracy is based on one trial, and we will add the results of 2 extra trials into the final version. As shown in Table 10, our SFPK outperforms various competitive baselines, including WF (Singh & Alistarh, 2020), STR (Kusupati et al., 2020), standard global magnitude pruning (Han et al., 2015), GMP (Gale et al., 2019), DNW (Wortsman et al., 2019; Molchanov et al., 2017b), and RIGL+ERK (Evci et al., 2020).

| ImageNet-1K | Gradual Unstructured Pruning | | | |
|---|---|---|---|---|
| ResNet-50 | Dense | Pruned | Rel. Drop | Sparsity |
| GMP | 76.69 | 70.59 | -7.95 | 95.00 |
| VD | 76.69 | 69.41 | -9.49 | 94.92 |
| VD | 76.69 | 71.81 | -6.36 | 94.94 |
| RIGL + ERK | 76.80 | 70.00 | -8.85 | 95.00 |
| DNW | 77.01 | 68.30 | -11.31 | 95.00 |
| STR | 77.01 | 70.97 | -7.84 | 94.80 |
| STR | 77.01 | 70.40 | -8.58 | 95.03 |
| Global Magnitude | 77.01 | 71.72 | -6.87 | 95.00 |
| WF | 77.01 | 72.12 | -6.30 | 95.00 |
| **SPFK (ours)** | **77.01** | **72.23** | **-6.21** | **95.00** |

Table 10: Unstructured gradual pruning with ResNet-50 on ImageNet-1K. For each experiment, we report the top-1 accuracy (%) of the pruned model and the relative performance drop (%) compared to the dense model.

**Diversity of SPFK mask particles.** We conduct additional experiments to study the diversity of the simulated particle distribution of SFPK. Specifically, at each intermediate step of SFPK, we track the mean and standard deviation of the loss function (the energy) and the average sparsity deviation of each mask particle. The average relative mask deviation of the $i$-th mask is computed by $1/(n-1) \cdot \sum_j \|\mathbf{m}_t^i - \mathbf{m}_t^j\|_2^2/\|\mathbf{m}_t^i\|_2^2$. We visualize the mean and standard deviation (across 10 mask particles) of both the loss value and the relative mask deviation.

As shown in Figure 5, as sparsity increases, the energy of each mask increases without drastic explosion. Meanwhile, the diversity of the masks progressively grows. This suggests that the SFPK mask distribution does not collapse and it generates diverse yet effective mask particles as sparsity decreases, allowing us to sample performant masks at the desired sparsity level.

**Ablation study on the ensemble effect in SFPK.** To identify how the ensemble scheme contributes to the performance gain of SFPK, we compare SFPK-pruner with an ensemble-based pruning method, termed SWAMP (Choi et al., 2024). We compare non-ensemble SFPK (SFPK w/o ens) with SWAMP on WRN32x4 and ResNet-50 using the same gradual pruning protocol as in Table 2, but with particle number $n = 1$. We quote the SWAMP results from the last row of Table 2 and the first row of Table 3 in (Choi et al., 2024). The results are shown in Table 11 and 12. We observe that SFPK works well even with only ONE particle, showing that both the SFPK update and particle simulation scheme contribute to the good performance of SFPK.

Table 12 also shows SFPK w/o ens still matches or exceeds SWAMP's performance on ResNet-50 ImageNet (IN-Valid) with less FLOPS. At 90% sparsity, SFPK achieves +3.4% accuracy increase compared to SWAMP within only ×0.05 training epochs. Hence, both the SFPK update (Proposition

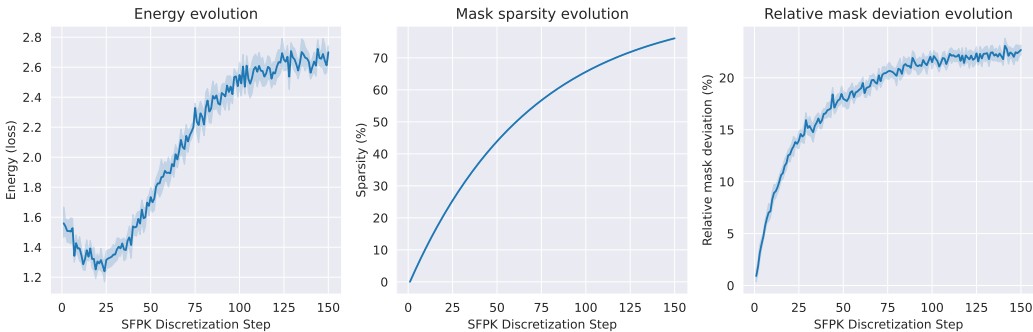

Figure 5: Visualization of the relative mask deviation among the SFPK mask particles. A larger deviation implies a higher degree of diversity of the SFPK masks.

2) and the particle simulation method (Section 4.2) contribute non-trivially to the benefits of SFPK. Our SFPK neither alters weights nor uses weight averaging, and it enjoys provable convergence (Proposition 3). In contrast, SWAMP is a combination of $L_1$ pruning and a weight averaging trick, requiring more training epochs to converge.

| SFPK (ours) v.s. SWAMP, WRN-32x4 (78.86) on CIFAR-100 | | | | |
|---|---|---|---|---|
| Method | Sparsity | Accuracy | Sparsity | Accuracy |
| SWAMP | 50% | 77.29 (0.53) | 90% | 77.14 (0.33) |
| SFPK w/o ens | 50% | 78.04 (0.09) | 90% | 77.38 (0.14) |
| SFPK (ours) | 50% | **78.35 (0.11)** | 90% | **77.61 (0.21)** |
| SWAMP | 75% | 77.35 (0.39) | 95% | 76.48 (0.73) |
| SFPK w/o ens | 75% | 77.68 (0.17) | 95% | 76.68 (0.19) |
| SFPK (ours) | 75% | **77.90 (0.24)** | 95% | **77.04 (0.26)** |

Table 11: 'SFPK w/o ens' refers to disabling the ensembles in SFPK by setting the particle number $n = 1$. Conducted on WRN-32x4 on the CIFAR-100 dataset.

**Comparison with $L_1$-Spred.** We compare our SFPK against the state-of-the-art $L_1$ optimizer, $L_1$-Spred (Ziyin & Wang, 2023), showing that our method consistently outperforms $L_1$-Spred on both CIFAR-10/100 and ImageNet. We use the same SFPK configuration as in Table 2. On CIFAR-10/100, we apply SFPK to prune the `timm` pretrained ResNet-18 weights to 99% sparsity over 15 pruning shots with a total 150 retrain epochs and report the accuracy of all sparsity checkpoints. We use SFPK to prune the same checkpoint as that in (Ziyin & Wang, 2023), `torchvision` ResNet-50 V2 pretrained weights, to 90% sparsity over 10 pruning shots and 100 retrain epochs. We report the accuracy of the 80% / 90% sparsity checkpoints and quote the best results from the CIFAR / ImageNet forms in the official Github repo of (Ziyin & Wang, 2023). As shown in Figure 6 and Table 13, our SFPK consistently outperforms $L_1$-Spred across different pruning settings.

**Comparison with 500-epoch RigL.** For a fair comparison, we reported the results of the 100-epoch RigL+ERK (Evci et al., 2020) in the Section 5, rather than the 500-epoch RigL (+ERK). To ensure fair comparisons between pruning and sparse-training methods, we conduct controlled experiments, with the same 100-epoch retraining, sparsity constraints, and pretrained weights. That's why we reported the 100-epoch RigL+ERK results, the best result among the 100-epoch RigL variants (Evci et al., 2020, Figure 2).

Table 14 compares SFPK with the 500-epoch RigL in accuracy and FLOPs. Due to the limited computation resource, we apply SFPK to the `torchivision` ResNet-50 V2 checkpoint to compensate for the fewer training epochs. The results show that, in practice, where a pretrained model is usually accessible, our SFPK outperforms the 500-epoch RigL in both accuracy and efficiency.

**Comparison with MEST.** Our paper focuses on a fair comparison within a post-training pruning (PTP) setting, e.g. pruning a pretrained dense model. We followed the PTP setting in (Singh &

| SFPK (ours) v.s. SWAMP, ResNet-50 (77.01) on ImageNet | | | | |
|---|---|---|---|---|
| Method | Sparsity | Accuracy | Epoch | Tra. FLOPs |
| SWAMP | 45.90% | 76.56 | 210 | 6.72 |
| SFPK w/o ens | 50.10% | 76.59 | 10 | 0.35 |
| SFPK (ours) | 50.10% | **76.82** | 10 | 0.35 |
| SWAMP | 68.80% | 76.51 | 330 | 10.56 |
| SFPK w/o ens | 68.40% | 76.38 | 20 | 0.55 |
| SFPK (ours) | 68.40% | **76.59** | 20 | 0.55 |
| SWAMP | 80.00% | 75.69 | 450 | 14.4 |
| SFPK w/o ens | 80.00% | 75.96 | 40 | 0.8 |
| SFPK (ours) | 80.00% | **76.18** | 40 | 0.8 |
| SWAMP | 86.00% | 74.25 | 570 | 18.24 |
| SFPK w/o ens | 87.41% | 75.19 | 65 | 0.99 |
| SFPK (ours) | 87.41% | **75.65** | 65 | 0.99 |
| SWAMP | 88.90% | 71.81 | 630 | 20.16 |
| SFPK w/o ens | 90.00% | 74.53 | 100 | 1.19 |
| SFPK (ours) | 90.00% | **75.24** | 100 | 1.19 |

Table 12: 'SFPK w/o ens' refers to disabling the ensembles in SFPK by setting the particle number $n = 1$. Conducted on ResNet-50 on the ImageNet dataset. We reported the training FLOPs in units of $1e18$ and inference FLOPs in units $1e9$

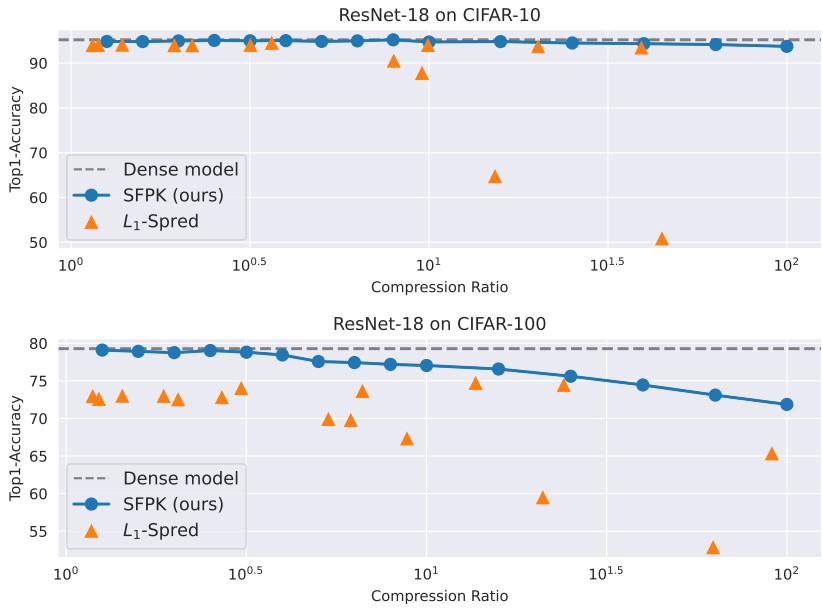

Figure 6: SFPK (ours) v.s. $L_1$-Spred on ResNet-18 CIFAR-10/100.

Alistarh, 2020) to compare pruning methods via controlled experiments, with the same retraining budgets, sparsity constraints, and pretrained checkpoints. Some sparse-training baselines were not included because the PTP setting differs from the sparse-training setting. PTP focuses on reducing the size of an off-the-shelf pretrained model and inference FLOPs. In contrast, sparse-training methods aim to reduce the pretraining memory footprint and training FLOPs. They start from sparsely and randomly initialized weights and require more training epochs than the retraining epochs in PTP.

We make a direct comparison with MEST (Yuan et al., 2021), a state-of-the-art sparse training method, using the same settings in Table 2. For CIFAR-10/100, we use unstructured SFPK to

| SFPK (ours) v.s. $L_1$-spred, ResNet-50 (V2) ImageNet | | | | |
|---|---|---|---|---|
| Method | Sparsity | Accuracy | Sparsity | Accuracy |
| $L_1$-spred + ft | 80% | 77.52 | 90% | 69.49 |
| SFPK+ (ours) | 80% | **77.97** | 90% | **76.56** |

Table 13: Both methods use the `torchvision` ResNet-50 V2 pretrained checkpoint with 80.34% accuracy.

| SFPK (ours) v.s. RigL, ResNet-50 (77.01) on ImageNet | | | | | |
|---|---|---|---|---|---|
| Method | Sparsity | Accuracy | Epoch | Tra. FLOPs | Inf. FLOPs |
| RigL+ERK | 80% | 75.1 | 100 | 1.34 | 3.44 |
| RigL+ERK 5× | 80% | 77.1 | 500 | 6.69 | 3.44 |
| SFPK (ours) | 80% | 76.18 | 40 | 0.80 | 2.62 |
| SFPK+ (ours) | 80% | **77.97** | 40 | 0.80 | 2.64 |
| RigL+ERK | 90% | 73.0 | 100 | 0.80 | 1.96 |
| RigL+ERK 5× | 90% | 76.4 | 500 | 3.94 | 1.96 |
| SFPK (ours) | 90% | 75.24 | 100 | 1.19 | 1.53 |
| SFPK+ (ours) | 90% | **76.56** | 100 | 1.19 | 1.53 |

Table 14: 'SFPK+' refers to applying SFPK to the `torchvision` ResNet-50 V2 pretrained checkpoint with 80.34% accuracy. We reported the training FLOPs in units of $1e18$ and inference FLOPs in units $1e9$.

prune pretrained ResNet-32 to 90%, 95%, and 98% sparsity, then retrain it for 150 epochs. For ImageNet, we apply gradual unstructured SFPK pruning to pretrained ResNet-50 to 90% sparsity with a total of 100 retraining epochs and 10 pruning shots. Then, we report the accuracy of the 80% sparsity checkpoint achieved at the 40-th epoch and the final 90% sparsity checkpoint. For a fair comparison between our 100-epoch SFPK and the 250-epoch MEST, we use a `torchvision` ResNet-50 V2 pretrained weight to compensate for the fewer training epochs. To ablate the effect of this checkpoint, we also implement a 100-epoch MEST that is initialized from the same ResNet-50 V2 checkpoint. We denote it by '100-epoch MEST+'. We quote the MEST results from the last row of Table 1 and Table 2 in (Yuan et al., 2021).

As shown in Table 15 and Table 16, our SFPK consistently outperforms MEST on CIFAR-10/100. Our 100-epoch SFPK consistently outperforms 150-epoch MEST on ImageNet. When using the `torchvision` ResNet-50 V2 pretrained weight, our 100-epoch SFPK again beats both the 250-epoch MEST and 100-epoch MEST+. We also compared their training and inference FLOPs. These results show that it is more beneficial to use our SFPK especially when well-trained checkpoints are accessible.

| ResNet-32 | CIFAR-10 (Dense Acc.: 94.78) | | CIFAR-100 (Dense Acc.: 74.53) | |
|---|---|---|---|---|
| Method | Sparsity | Accuracy | Sparsity | Accuracy |
| MEST+EM&S | 90% | 93.27 (0.14) | 90% | 71.30 (0.31) |
| SFPK (ours) | 90% | **94.57 (0.09)** | 90% | **72.67 (0.12)** |
| MEST+EM&S | 95% | 92.44 (0.13) | 95% | 70.36 (0.05) |
| SFPK (ours) | 95% | **94.05 (0.17)** | 95% | **71.69 (0.15)** |
| MEST+EM&S | 98% | 90.51 (0.11) | 98% | 67.16 (0.25) |
| SFPK (ours) | 98% | **92.42 (0.20)** | 98% | **67.48 (0.19)** |

Table 15: SFPK (ours) v.s. MEST on ResNet-32 CIFAR-10/100.

| SFPK (ours) v.s. MEST, ResNet50 (77.01) on ImageNet | | | | | |
|---|---|---|---|---|---|
| Method | Sparsity | Accuracy | Epochs | Tra. FLOPs | Inf. FLOPs |
| MEST ×0.5 | 80% | 75.39 | 75 | 0.74 | 1.70 |
| MEST ×1 | 80% | 75.75 | 150 | 1.27 | 1.70 |
| MEST ×1.7 | 80% | 77.19 | 250 | 2.15 | 1.70 |
| MEST+ ×1 | 80% | 75.99 | 150 | 1.27 | 1.70 |
| SFPK (ours) | 80% | 76.18 | 40 | 0.80 | 2.62 |
| SFPK+ (ours) | 80% | **77.97** | 40 | 0.80 | 2.64 |
| MEST ×0.5 | 90% | 72.58 | 75 | 0.39 | 0.90 |
| MEST ×1 | 90% | 75.1 | 150 | 0.60 | 0.90 |
| MEST ×1.7 | 90% | 76.13 | 250 | 1.11 | 0.90 |
| SFPK (ours) | 90% | 75.24 | 100 | 1.19 | 1.53 |
| SFPK+ (ours) | 90% | **76.56** | 100 | 1.19 | 1.53 |

Table 16: 'SFPK+'/'MEST+' refers to applying SFPK/MEST to the `torchvision` ResNet-50 V2 pretrained checkpoint with $80.34\%$ accuracy. We reported the training FLOPs in units of $1e18$ and inference FLOPs in units $1e9$.

# D  THEORETICAL RESULTS

## D.1  PROOF DETAILS

**Definition 3** (Weak Convergence). Let $\mu$ be the Lebesgue measure on $\mathbb{R}^d$ and $\Omega \subset \mathbb{R}^d$ be a $\nu$-measurable set. A bounded sequence of distributions $\{\pi^n\}_{n \geqslant 1} \subset \mathcal{P}(\Omega)$ is said to **converge weakly** to $\pi \in \mathcal{P}(\Omega)$, denoted by $\pi^n \rightharpoonup \pi$, if and only if $\langle \pi^n, \phi \rangle \to \langle \pi, \phi \rangle$ as $n \to \infty$ holds for all bounded and smooth test functions $\phi(\cdot)$ on $\Omega$.

**Proposition 4** (Existence and Uniqueness of the Optimal Mask Distribution (formal version of Proposition 1)). *Let $s(\cdot) : \mathbb{R}^d \mapsto \mathbb{R}_+$ be the soft sparsity function, $L_\varepsilon(\cdot) : \mathbb{R}^d \mapsto \mathbb{R}_+$ be the model loss function that is a.e. smooth, $\kappa(\cdot, \cdot) : \mathbb{R}^d \times \mathbb{R}^d \mapsto \mathbb{R}_+$ denotes the RBF function kernel, we consider the following distributional optimization problem*

$$\min_{\mathcal{P}(M_t^\rho)} \langle L_\varepsilon, \pi \rangle + \lambda \mathcal{R}[\pi] \triangleq \int_{M_t} L_\varepsilon(\mathbf{m}) \pi(\mathrm{d}\mathbf{m}) + \frac{\lambda}{2} \int_{M_t} \kappa(\mathbf{m}, \mathbf{m}') \pi(\mathrm{d}\mathbf{m}) \pi(\mathrm{d}\mathbf{m}'), \qquad (9)$$

*with $\lambda > 0$ and $M_t^\rho \triangleq \left\{ \mathbf{m} \in \mathbb{R}^d : s(\mathbf{m}) \in [t, t + \rho] \right\}$. For simplicity, we denote the functional objective as $\mathcal{L}[\cdot] \triangleq \langle L_\varepsilon, \cdot \rangle + \lambda \mathcal{R}[\cdot]$. Assume that $L_\varepsilon(\cdot)$ satisfies the following assumptions:*

*(A1).* **Strict sparsity monotonicity**: *the optimal loss at lower sparsity levels is always strictly smaller than at the higher sparsity levels. Specifically, there exists $\alpha > 0$, for any sparsity levels $t < t'$, such that $\inf_{\mathbf{m} \in M_t^0} L_\varepsilon(\mathbf{m}) < \inf_{\mathbf{m} \in M_{t'}^0} L_\varepsilon(\mathbf{m}) - \alpha(t' - t)^2$.*

*(A2).* **Lower semi-continuity**: *for all $\mathbf{m} \in \mathbb{R}^d$, for any sequence $\{\mathbf{m}^i\}_{i \geqslant 1}$ that converges to $\mathbf{m}$, it holds that $\liminf_{i \to \infty} L_\varepsilon(\mathbf{m}^i) \geqslant L_\varepsilon(\mathbf{m})$.*

*Then, for any sparsity level $t$, there exists a $\pi_t$ such that $\mathcal{L}(\pi_t) = \inf \{\mathcal{L}(\pi) : \pi \in \mathcal{P}(M_t^\rho)\}$. Moreover, the global minimizer of $\mathcal{L}[\cdot]$ in $\mathcal{P}(M_t^\rho)$ is unique.*

*Proof of Proposition 4.* By definition, the objective functional is lower bounded by $0$, which implies $0 \leqslant \mathcal{L}^* \triangleq \inf \{\mathcal{L}[\pi] : \pi \in \mathcal{P}(M_t^\rho)\} < \infty$. Thus, there exists a sequence $\{\mathcal{L}_n\}_{n \geqslant 1}$ in $\mathcal{L}[\mathcal{P}(M_t^\rho)]$, satisfying $\mathcal{L}_n \to \mathcal{L}^*$ as $n \to \infty$ and for each $n$, $\exists \pi^n \in \mathcal{P}(M_t^\rho)$ such that $\mathcal{L}_n = \mathcal{L}[\pi^n]$. Then we want to show the sequence $\{\pi^n\}_{n \geqslant 1}$ is **tight**, that is, for any $\epsilon > 0$, there exists a compact set $K \subset M_t^\rho$ such that for any $n \in \mathbb{N}$, we have $\pi^n(K) > 1 - \epsilon$.

Once the tightness of $\{\pi^n\}_{n \geqslant 1}$ is established, we can apply the Prokhorov's theorem (Billingsley, 1968, Theorem 5.1) to extract a subsequence $\{\pi^{n_k}\}_{k \geqslant 1}$ of $\{\pi^n\}_{n \geqslant 1}$ and a distribution $\pi_t \in \mathcal{P}(M_t^\rho)$ such that $\pi^{n_k} \rightharpoonup \pi_t$ for $k \to \infty$. According to (Ambrosio et al., 2006, Lemma 5.1.7), (A2) implies

the lower semi-continuity of the linear functional $\langle L_\varepsilon, \cdot \rangle$. Furthermore, the continuity of $\kappa(\cdot, \cdot)$ implies that $\mathcal{L}[\cdot]$ is also lower semi-continuous, which further yields

$$\mathcal{L}[\pi_t] \leqslant \liminf_{k \to \infty} \mathcal{L}[\pi^{n_k}] = \liminf_{k \to \infty} \mathcal{L}_{n_k} = \lim_{n \to \infty} \mathcal{L}_n = \mathcal{L}^*. \tag{10}$$

By definition of $\mathcal{L}^*$, it holds that $\mathcal{L}^* \leqslant \mathcal{L}[\pi_t] \leqslant \mathcal{L}^*$, showing that the infimum is indeed achievable in $\mathcal{P}(M_t^\rho)$ at $\pi_t$, i.e. $\mathcal{L}[\pi_t] = \mathcal{L}^*$. This proves the existence of the optimal mask distribution.

Then we prove the uniqueness of $\pi_t$ by contradiction. Suppose $\mathcal{L}^*$ is attained at two different distributions $\pi_t^1$ and $\pi_t^2$. By (Arbel et al., 2019, Lemma 25), the strict convexity of $\kappa(\cdot, \cdot)$ implies the strict convexity of $\mathcal{R}[\cdot]$ and $\mathcal{L}[\cdot]$. Let $\beta \in (0, 1)$ and $\pi_t^3 = \beta \pi_t^1 + (1 - \beta)\pi_t^2$, it holds that $\pi_t^3 \in \mathcal{P}(M_t^\rho)$ and

$$\mathcal{L}[\pi_t^3] < \beta \mathcal{L}[\pi_t^1] + (1 - \beta)\mathcal{L}[\pi_t^2] = \mathcal{L}^*. \tag{11}$$

Thus, this contradicts the definition of $\mathcal{L}^*$, which implies that $\mathcal{L}^*$ is uniquely attained at $\pi_t$.

To complete the proof, we only need to establish the tightness of the aforementioned sequence $\{\pi^n\}_{n \geqslant 1}$. This can be proved via contradiction. Suppose $\{\pi^n\}_{n \geqslant 1}$ is not tight, then there exists $\epsilon > 0$ such that for all $k \in \mathbb{N}$, there exist $n_k \in N$, for arbitrary $I_k \subset M_t^\rho$ with $\pi^{n_k}(I_k) \leqslant 1 - \epsilon$. This implies

$$\mathcal{L}_{n_k} = \mathcal{L}[\pi^{n_k}] = \int_{I_k} + \int_{M_t^\rho \setminus I_k} L_\varepsilon(\mathbf{m}) \pi^{n_k}(\mathrm{d}\mathbf{m}) + \lambda \mathcal{R}[\pi^{n_k}] \tag{12}$$

$$\geqslant \int_{I_k} + \int_{M_t^\rho \setminus I_k} L_\varepsilon(\mathbf{m}) \pi^{n_k}(\mathrm{d}\mathbf{m}) \tag{13}$$

$$\geqslant \inf\{\mathcal{L}[\pi] : \pi \in \mathcal{P}(I_k)\} + \pi^{n_k}(M_t^\rho \setminus I_k) \cdot \inf\{L_\varepsilon(\mathbf{m}) : \mathbf{m} \in M_t^\rho \setminus I_k\} \tag{14}$$

$$\geqslant \inf\{\mathcal{L}[\pi] : \pi \in \mathcal{P}(I_k)\} + \epsilon \inf\{L_\varepsilon(\mathbf{m}) : \mathbf{m} \in M_t^\rho \setminus I_k\}. \tag{15}$$

Let $I_k \triangleq M_t^{\rho - \frac{1}{k}}$, it holds that $M_t^\rho \setminus I_k = M_{t + \rho - \frac{1}{k}}^{\frac{1}{k}}$. Therefore, the condition (A1) implies that

$$\mathcal{L}_{n_k} \geqslant \inf\{\mathcal{L}[\pi] : \pi \in \mathcal{P}(I_k)\} + \epsilon \inf\{L_\varepsilon(\mathbf{m}) : \mathbf{m} \in M_t^\rho \setminus I_k\} \tag{16}$$

$$= \inf\{\mathcal{L}[\pi] : \pi \in \mathcal{P}(M_t^{\rho - \frac{1}{k}})\} + \epsilon \inf\left\{L_\varepsilon(\mathbf{m}) : \mathbf{m} \in M_{t + \rho - \frac{1}{k}}^{\frac{1}{k}}\right\} \tag{17}$$

$$> \inf\{\mathcal{L}[\pi] : \pi \in \mathcal{P}(M_t^{\rho - \frac{1}{k}})\} + \epsilon \left(\inf\left\{L_\varepsilon(\mathbf{m}) : \mathbf{m} \in M_t^{\frac{1}{k}}\right\} + \alpha\left(\rho - 1/k\right)^2\right) \tag{18}$$

$$\geqslant \inf\{\mathcal{L}[\pi] : \pi \in \mathcal{P}(M_t^{\rho - \frac{1}{k}})\} + \epsilon\alpha(\rho - 1/k)^2. \tag{19}$$

By taking the limitation $k \to \infty$ on both sides, it holds that $\mathcal{L}^* \geqslant \mathcal{L}^* + \epsilon\alpha\rho^2$, with $\epsilon\alpha\rho^2 > 0$, which is a contradiction. Thus, this implies the tightness of $\{\pi^n\}_{n \geqslant 1}$ and completes the proof. $\square$

**Proposition 5** (Optimal Local Distributional Transition (formal version of Proposition 2)). *Following the notations and conditions introduced in Proposition 4, let $\pi_t$ be the optimal mask distribution at sparsity level $t$, let $\mu_{\pi_t}(\cdot) \triangleq \int \kappa(\cdot, \mathbf{m})\pi_t(\mathrm{d}\mathbf{m})$ be the $\kappa$-mean embedding of $\pi_t$, we consider the following local distributional transition problem*

$$\min_\nu \widehat{\mathcal{L}}[\nu] \triangleq \langle L_\varepsilon + \lambda\mu_{\pi_t}, \nu * \pi_t - \pi_t \rangle, \text{ s.t. } \nu \in \widehat{P}\left(M_{t+\Delta t}^\rho\right) \tag{20}$$

$$\widehat{\mathcal{P}}\left(M_{t+\Delta t}^\rho\right) \triangleq \{\nu * \pi_t \mid \mathrm{supp}(\nu(\cdot|\mathbf{m})) \in B_{r_t\Delta t}, \forall \mathbf{m} \in M_t^\rho\} \cap \mathcal{P}\left(M_{t+\Delta t}^\rho\right), \tag{21}$$

*where $r_t > 0$ is the local radius satisfying $r_t\|\nabla s(\mathbf{m})\|_2 > 1$ for all $\mathbf{m} \in M_t^\rho$. We further assume that $s(\cdot)$ satisfies*

*(A3). **Local regularity:** $s(\cdot)$ provides a local norm control. Specifically, for any $\mathbf{m}$, there exists $\epsilon > 0$, and $C > 0$, such that any $\mathbf{m}'$ satisfying $\|\mathbf{m}' - \mathbf{m}\|_2 \leqslant \epsilon$ holds that*
$$\|\mathbf{m}' - \mathbf{m}\|_2 \leqslant C|s(\mathbf{m}') - s(\mathbf{m})|.$$

*Then, $\nu_t(\cdot|\mathbf{m}) \triangleq \mathrm{Dirac}(\cdot; T(\mathbf{m}; \pi_t)\Delta t)$ achieves the optimal value of (20) within an error of order $\mathcal{O}(\Delta t^2)$, that is, $\hat{\mathcal{L}}[\nu_t] - \widehat{\mathcal{L}}^* = \mathcal{O}(\Delta t^2)$ and $\nu_t \in \widehat{\mathcal{P}}(M_{t+\Delta t-C\Delta t^2}^\rho)$ for some constant $C$. Specifically, let $\mathbf{g} \triangleq -\nabla L_\varepsilon(\mathbf{m}) - \lambda\nabla\mu_{\pi_t}(\mathbf{m})$ and $\mathbf{s} \triangleq \nabla s(\mathbf{m})$, $T(\mathbf{m}; \pi_t)$ is given by*

$$T(\mathbf{m}; \pi_t) \triangleq \frac{\mathbf{s}}{\|\mathbf{s}\|_2^2} + \left(\frac{r_t^2\|\mathbf{s}\|_2^2 - 1}{\|\mathbf{s}\|_2^2\|\mathbf{g}\|_2^2 - (\mathbf{s}^\top\mathbf{g})^2}\right)^{\frac{1}{2}} \cdot \left(\mathbf{I} - \frac{\mathbf{s}\mathbf{s}^\top}{\|\mathbf{s}\|_2^2}\right)\mathbf{g}. \tag{22}$$

*Proof of Proposition 5.* For simplicity, we denote $F(\cdot) \triangleq L_\varepsilon(\cdot) + \lambda\mu_{\pi_t}(\cdot)$. By standard derivation, it holds that

$$\widehat{\mathcal{L}}[\nu] = \int F(\mathbf{m})((\nu * \pi_t)(\mathbf{m}) - \pi_t(\mathbf{m}))\mathrm{d}\mathbf{m} \tag{23}$$

$$= \int F(\mathbf{m}) \left( \int \nu(\boldsymbol{\delta}|\mathbf{m} - \boldsymbol{\delta})\pi_t(\mathbf{m} - \boldsymbol{\delta})\mathrm{d}\boldsymbol{\delta} - \pi_t(\mathbf{m}) \right) \mathrm{d}\mathbf{m} \tag{24}$$

$$= \int F(\mathbf{m}) \int (\nu(\boldsymbol{\delta}|\mathbf{m} - \boldsymbol{\delta})\pi_t(\mathbf{m} - \boldsymbol{\delta}) - \nu(\boldsymbol{\delta}|\mathbf{m})\pi_t(\mathbf{m})) \, \mathrm{d}\boldsymbol{\delta}\mathrm{d}\mathbf{m} \tag{25}$$

$$= - \iint F(\mathbf{m}) \left( \nabla_{\mathbf{m}}(\nu(\boldsymbol{\delta}|\mathbf{m})\pi_t(\mathbf{m}))^\top \boldsymbol{\delta} + \mathcal{O}(\|\boldsymbol{\delta}\|_2^2) \right) \mathrm{d}\mathbf{m}\mathrm{d}\boldsymbol{\delta} \tag{26}$$

$$= - \iint F(\mathbf{m})\nabla_{\mathbf{m}}(\nu(\boldsymbol{\delta}|\mathbf{m})\pi_t(\mathbf{m}))^\top \boldsymbol{\delta}\mathrm{d}\mathbf{m}\mathrm{d}\boldsymbol{\delta} + \mathcal{O}(\Delta t^2) \tag{27}$$

$$= \iint \left( \nabla F(\mathbf{m})^\top \boldsymbol{\delta} \right) \nu(\mathrm{d}\boldsymbol{\delta}|\mathbf{m})\pi_t(\mathrm{d}\mathbf{m}) + \mathcal{O}(\Delta t^2). \tag{28}$$

To identify a minimizer of $\widehat{\mathcal{L}}[\nu]$ within $\mathcal{O}(\Delta t^2)$ error, one can resort to solving (28) in a point-wise (microscopic) manner. Specifically, we consider the following deterministic optimization problem

$$\min_{\boldsymbol{\delta}} \nabla F(\mathbf{m})^\top \boldsymbol{\delta}, \text{ s.t. } s(\mathbf{m} + \boldsymbol{\delta}) = s(\mathbf{m}) + \Delta t, \ \|\boldsymbol{\delta}\|_2 \leqslant r_t\Delta t. \tag{29}$$

Suppose $\boldsymbol{\delta}^*(\mathbf{m})$ is the minimizer of (29), let $\nu^*(\cdot|\mathbf{m})$ denotes the minimizer of the first term in (20), then it holds that $\nu^*(\cdot|\mathbf{m}) = \mathrm{Dirac}(\cdot; \boldsymbol{\delta}^*(\mathbf{m}))$. This can be proved by definition: for any $\mathbf{m} \in M_t^\rho$, suppose $A(\mathbf{m})$ is the support of $\nu^*(\cdot|\mathbf{m})$ such that $\forall \boldsymbol{\delta}' \in A, \|\boldsymbol{\delta}'\|_2 \leqslant r_t\Delta t, s(\mathbf{m}+\boldsymbol{\delta}') = s(\mathbf{m})+\Delta t$, then it holds that

$$\iint \left( \nabla F(\mathbf{m})^\top \boldsymbol{\delta} \right) \nu^*(\mathrm{d}\boldsymbol{\delta}|\mathbf{m})\pi_t(\mathrm{d}\mathbf{m}) \tag{30}$$

$$= \int \left( \int_{A(\mathbf{m})} + \int_{A(\mathbf{m})^c} \left( \nabla F(\mathbf{m})^\top \boldsymbol{\delta} \right) \nu^*(\mathrm{d}\boldsymbol{\delta}|\mathbf{m}) \right) \pi_t(\mathrm{d}\mathbf{m}) \tag{31}$$

$$\leqslant \int \left( \nabla F(\mathbf{m})^\top \boldsymbol{\delta}^*(\mathbf{m}) \right) \nu^*(A(\mathbf{m})|\mathbf{m})\pi_t(\mathrm{d}\mathbf{m}) + 0 \tag{32}$$

$$= \int \left( \nabla F(\mathbf{m})^\top \boldsymbol{\delta}^*(\mathbf{m}) \right) \pi_t(\mathrm{d}\mathbf{m}) \tag{33}$$

$$= \iint \left( \nabla F(\mathbf{m})^\top \boldsymbol{\delta} \right) \mathrm{Dirac}(\mathrm{d}\boldsymbol{\delta}; \boldsymbol{\delta}^*(\mathbf{m}))\pi_t(\mathrm{d}\mathbf{m}). \tag{34}$$

Notice that solving $\boldsymbol{\delta}^*(\mathbf{m})$ from (29) is still challenging, since the constraint is nonlinear. Thus, we aim to slightly relax the constraint within an $\mathcal{O}(\Delta t^2)$-order error. Specifically, we consider the following convex optimization problem

$$\min_{\boldsymbol{\delta}} \nabla F(\mathbf{m})^\top \boldsymbol{\delta}, \text{ s.t. } \nabla s(\mathbf{m})^\top \boldsymbol{\delta} = \Delta t, \ \|\boldsymbol{\delta}\|_2 \leqslant r_t\Delta t. \tag{35}$$

Clearly, (A3) implies that the minimizer of (35), denoted by $\widehat{\boldsymbol{\delta}}^*(\mathbf{m})$, achieves the optimum of (29) within a $O(\Delta t^2)$-order error. Thus, one can establish an $O(\Delta t^2)$-order error approximation for $\nu^*(\cdot; \mathbf{m})$ as $\mathrm{Dirac}(\cdot; \widehat{\boldsymbol{\delta}}^*(\mathbf{m}))$. The final step to finishing the proof is to derive $\widehat{\boldsymbol{\delta}}^*(\mathbf{m})$. As (35) is a convex optimization problem, the optimum is attained at the decision boundary. Moreover, the optimum lies in the linear space spanned by $\{\nabla F(\mathbf{m}), \nabla s(\mathbf{m})\}$. For simplicity, we denote $\mathbf{g} \triangleq -\nabla F(\mathbf{m})$ and $\mathbf{s} \triangleq \nabla s(\mathbf{m})$, then we parameterized the optimum of (35) as $a\mathbf{g} + b\mathbf{s}$. Finally, the proof is completed by solving the coefficients $a$ and $b$ from the following equations

$$\mathbf{s}^\top (a\mathbf{g} + b\mathbf{s}) = \Delta t, \quad \|a\mathbf{g} + b\mathbf{s}\|_2 = r_t\Delta t, \tag{36}$$

which yields

$$a = \left( \frac{r_t^2\|\mathbf{g}\|_2^2 - 1}{\|\mathbf{g}\|_2^2\|\mathbf{s}\|_2^2 - (\mathbf{g}^\top\mathbf{s})} \right)^{\frac{1}{2}} \cdot \Delta t, \quad b = \frac{\left( \Delta t - \left( \mathbf{g}^\top\mathbf{s} \right) a \right)}{\|\mathbf{g}\|_2^2}. \tag{37}$$

Therefore, the proof is completed by taking

$$T(\mathbf{m};\pi_t) \triangleq \widehat{\boldsymbol{\delta}}^*(\mathbf{m}) = a\mathbf{g} + b\mathbf{s} = \frac{\mathbf{s}}{\|\mathbf{s}\|_2^2} + \left(\frac{r_t^2\|\mathbf{s}\|_2^2 - 1}{\|\mathbf{s}\|_2^2\|\mathbf{g}\|_2^2 - (\mathbf{s}^\top\mathbf{g})^2}\right)^{\frac{1}{2}} \cdot \left(\mathbf{I} - \frac{\mathbf{s}\mathbf{s}^\top}{\|\mathbf{s}\|_2^2}\right)\mathbf{g}. \quad (38)$$

$\square$

**Proposition 6** (Derivation of SFPK (formal version of Proposition 3))**.** *Following the previously introduced notations and conditions, we further assume that*

*(A4).* **Quadratic growth***: the growth of expected loss is quadratically bounded w.r.t sparsity levels. Specifically, there exists $\beta > 0$, for any sparsity levels $t < t'$, such that*

$$\inf_{\mathbf{m}\in M_{t'}^0} L_\varepsilon(\mathbf{m}) < \inf_{\mathbf{m}\in M_t^0} L_\varepsilon(\mathbf{m}) + \beta(t'-t)^2.$$

*Then, by taking $\Delta t \to 0$, the sequence $\{\widetilde{\pi}_{k\Delta t}\}$ constructed by $\widetilde{\pi}_{t+\Delta t} \triangleq \nu_t * \widetilde{\pi}_t$ converges weakly to the Sparsity Evolutionary Fokker-Planck-Kolmogorov Equation (SFPK), denoted by*

$$\partial_t \pi_t = -\nabla \cdot [T(\cdot;\pi_t)\pi_t], \quad (39)$$

*where $\nabla \cdot [\cdot]$ denotes the divergence operator and $T(\cdot;\pi_t)$ is the optimal local transition function defined in Proposition 5.*

*Proof of Proposition 6.* The proof sketch is summarized as follows: 1) we first show that the sequence $\{\widetilde{\pi}_{k\Delta t}\}_k$ converges weakly to the distribution-valued differential equation $(\widetilde{\pi}_t)_t$ that satisfies the SFPK $\partial_t \widetilde{\pi}_t = -\nabla \cdot [T(\cdot;\widetilde{\pi}_t)\widetilde{\pi}_t]$, then 2) we show that $\widetilde{\pi}_t = \pi_t$ for any $t \in [0, 1 - d'/d]$ with $\pi_t$ denoting the optimal mask distribution defined in Proposition 4.

**Step 1.** To establish the weak convergence of $\{\pi_{k\Delta t}\}_k$, we need to show for any $t$ and $k \triangleq \lfloor t/\Delta t \rfloor$, it holds that

$$\frac{1}{\Delta t}(\widetilde{\pi}_{(k+1)\Delta t} - \widetilde{\pi}_{k\Delta t}) \rightharpoonup -\nabla \cdot [T(\cdot,\widetilde{\pi}_{k\Delta t})\widetilde{\pi}_{k\Delta t}] \quad (40)$$

as $\Delta t \to 0$. By definition, this is equivalent to

$$\lim_{\Delta t \to 0} \frac{1}{\Delta t} \int (\widetilde{\pi}_{(k+1)\Delta t}(\mathbf{m}) - \widetilde{\pi}_{k\Delta t}(\mathbf{m}))\phi(\mathbf{m})d\mathbf{m} = -\int \nabla_\mathbf{m} \cdot [T(\mathbf{m},\widetilde{\pi}_{k\Delta t})\widetilde{\pi}_{k\Delta t}(\mathbf{m})]\phi(\mathbf{m})d\mathbf{m}, \quad (41)$$

where $\phi(\cdot)$ is an arbitrary bounded smooth test function. Then, the desired weak convergence can be proved by showing that, for any $t$, let $k \triangleq \lfloor t/\Delta t \rfloor$, $\widetilde{\pi}_{k\Delta t} \rightharpoonup \widetilde{\pi}_t$ as $\Delta t \to 0$. According to Proposition 5, for any $k$, $\widetilde{\pi}_{(k+1)\Delta t} = \nu_{k\Delta t} * \widetilde{\pi}_{k\Delta t}$m which implies

$$\frac{1}{\Delta t}\int (\widetilde{\pi}_{(k+1)\Delta t}(\mathbf{m}) - \widetilde{\pi}_{k\Delta t}(\mathbf{m}))\phi(\mathbf{m})d\mathbf{m} \quad (42)$$

$$= \frac{1}{\Delta t}\int (\nu_{k\Delta t} * \widetilde{\pi}_{k\Delta t}(\mathbf{m}) - \widetilde{\pi}_{k\Delta t}(\mathbf{m}))\phi(\mathbf{m})d\mathbf{m} \quad (43)$$

$$= \frac{1}{\Delta t}\iint (\nu_{k\Delta t}(\boldsymbol{\delta}|\mathbf{m}-\boldsymbol{\delta})\widetilde{\pi}_{k\Delta t}(\mathbf{m}-\boldsymbol{\delta}) - \nu_{k\Delta t}(\boldsymbol{\delta}|\mathbf{m})\widetilde{\pi}_{k\Delta t}(\mathbf{m}))d\boldsymbol{\delta}\phi(\mathbf{m})d\mathbf{m} \quad (44)$$

$$= -\frac{1}{\Delta t}\iint \nabla_\mathbf{m}(\nu_{k\Delta t}(\boldsymbol{\delta}|\mathbf{m})\widetilde{\pi}_{k\Delta t}(\mathbf{m}))^\top \boldsymbol{\delta}\phi(\mathbf{m})d\boldsymbol{\delta}d\mathbf{m} + \mathcal{O}(\Delta t) \quad (45)$$

$$= \frac{1}{\Delta t}\iint \nabla\phi(\mathbf{m})^\top \boldsymbol{\delta}\nu_{k\Delta t}(\boldsymbol{\delta}|\mathbf{m})\widetilde{\pi}_{k\Delta t}(\mathbf{m})d\boldsymbol{\delta}d\mathbf{m} + \mathcal{O}(\Delta t) \quad (46)$$

$$= \frac{1}{\Delta t}\iint \nabla\phi(\mathbf{m})^\top \boldsymbol{\delta}\text{Dirac}(d\boldsymbol{\delta}; T(\mathbf{m},\widetilde{\pi}_{k\Delta t})\Delta t))\widetilde{\pi}_{k\Delta t}(\mathbf{m})d\mathbf{m} + \mathcal{O}(\Delta t) \quad (47)$$

$$= \int \nabla\phi(\mathbf{m})^\top T(\mathbf{m},\widetilde{\pi}_{k\Delta t})\widetilde{\pi}_{k\Delta t}(\mathbf{m})d\mathbf{m} + \mathcal{O}(\Delta t) \quad (48)$$

$$= -\int \nabla_\mathbf{m} \cdot [T(\mathbf{m},\widetilde{\pi}_{k\Delta t})\widetilde{\pi}_{k\Delta t}(\mathbf{m})]\phi(\mathbf{m})d\mathbf{m} + \mathcal{O}(\Delta t). \quad (49)$$

This implies the statement in (41). Then, we only need to show for $k \triangleq \lfloor t/\Delta t \rfloor$, $\widetilde{\pi}_{k\Delta t} \rightharpoonup \widetilde{\pi}_t$ as $\Delta t \to 0$. This equivalent to show that $(\widetilde{\pi}_{(k+1)\Delta t} - \widetilde{\pi}_{k\Delta t}) \rightharpoonup 0$ as $\Delta t \to 0$, which implies the

existence and the continuity (w.r.t the weak topology) of the limiting process $(\widetilde{\pi}_t)_t$. Recall that since $L_\varepsilon(\cdot)$, $\kappa(\cdot, \cdot)$ and $s(\cdot)$ are a.e.-smooth on the support of $\widetilde{\pi}_{k\Delta t}$, the right hand side of (41) is bounded, indicating that $\langle \widetilde{\pi}_{(k+1)\Delta t} - \widetilde{\pi}_{k\Delta t}, \phi \rangle = \mathcal{O}(\Delta t)$ holds for any bounded continuous $\phi(\cdot)$, which further implies $(\widetilde{\pi}_{(k+1)\Delta t} - \widetilde{\pi}_{k\Delta t}) \rightharpoonup 0$ as $\Delta t \to 0$.

**Step 2.** Suppose $\pi_t$ is the optimal mask distribution defined in Proposition 4. We aim to prove that if $\widetilde{\pi}_0 = \pi_0$, it holds that $\widetilde{\pi}_t = \pi_t$ for any $t \in [0, 1 - d'/d]$. This is equivalent to show $\langle \widetilde{\pi}_t - \pi_t, \phi \rangle = 0$ for any bounded continuous $\phi(\cdot)$ and any $t \in [0, 1 - d'/d]$. To this end, we study the evolution of the deviation term $|\langle \widetilde{\pi}_t - \pi_t, \phi \rangle|$ under a finite sparsity increment.

First, we show that $\pi_{t+\Delta t} \in \widehat{\mathcal{P}}(M_{t+\Delta t}^\rho)$, which implies there exist an oracle transition kernel $\nu_t^*(\cdot|\cdot)$ and $r_t > 0$ such that $\pi_{t+\Delta t} = \nu_t^* * \pi_t$ and $\mathrm{supp}(\nu_t^*(\cdot|\mathbf{m})) \subset B_{r_t \Delta t}$ for any $\mathbf{m} \in M_t^\rho$. Recall that the functional $\mathcal{L}[\cdot]$ is strictly convex, there exists $c > 0$, such that

$$\frac{c}{2}\|\pi_{t+\Delta t} - \pi_t\|_{L^2}^2 \leqslant \left\langle \frac{\delta}{\delta \pi_t}\mathcal{L}, \pi_t - \pi_{t+\Delta t} \right\rangle + \mathcal{L}[\pi_{t+\Delta t}] - \mathcal{L}[\pi_t] \overset{\text{Def.}}{\leqslant} \mathcal{L}[\pi_{t+\Delta t}] - \mathcal{L}[\pi_t] \overset{\text{(A4)}}{\leqslant} \beta \Delta t^2. \quad (50)$$

Therefore, by taking $r_t = 2\beta/c$, we have

$$\sup_\phi |\langle \pi_{t+\Delta t} - \pi_t, \phi \rangle| \leqslant \|\pi_{t+\Delta t} - \pi_t\|_{L^2}\|\phi\|_{L_\infty} \leqslant r_t \Delta t,$$

which indicates the 1-Wasserstein distance (Ambrosio et al., 2006) between $\pi_{t+\Delta t}$ and $\pi_t$ is bounded by $r_t \Delta t$ and hence $\pi_{t+\Delta t} \in \widehat{\mathcal{P}}(M_{t+\Delta t}^\rho)$, proving the existence of the oracle transition kernel $\nu_t^*$. Now, we can feel free to represent $\pi_{t+\Delta t}$ with $\nu_t^* * \pi_t$. For simplicity, we slightly abuse the notations and denote all the constants as $C$ and $C_t$ in the remainder of the paper. For any bounded smooth test function $\phi$, it holds that

$$|\langle \widetilde{\pi}_{t+\Delta t} - \pi_{t+\Delta t}, \phi \rangle| = |\langle \nu_t * \widetilde{\pi}_t - \nu_t^* * \pi_t, \phi \rangle| \quad (51)$$

$$\leqslant \underbrace{|\langle (\nu_t - \nu_t^*) * \pi_t, \phi \rangle|}_{\text{(T1): localization error}} + \underbrace{|\langle \nu_t * (\widetilde{\pi}_t - \pi_t), \phi \rangle|}_{\text{(T2): simulation error}}. \quad (52)$$

Then, we bound the terms (T1) and (T2) respectively. For any bounded smooth test function $\phi(\cdot)$ and an arbitrary small $\eta$, we denote $\Omega_\eta \triangleq \{\mathbf{m} : \phi(\mathbf{m}) > \eta/2\}$. By definition,

$$|\langle \nu_t * (\widetilde{\pi}_t - \pi_t), \phi \rangle| = \left|\iint \nu_t(\boldsymbol{\delta}|\mathbf{m}-\boldsymbol{\delta})(\widetilde{\pi}_t(\mathbf{m}-\boldsymbol{\delta}) - \pi_t(\mathbf{m}-\boldsymbol{\delta}))\phi(\mathbf{m})\mathrm{d}\boldsymbol{\delta}\mathrm{d}\mathbf{m}\right| \quad (53)$$

$$\leqslant \left|\iint_{\Omega_\eta} \nu_t(\boldsymbol{\delta}|\mathbf{m})(\widetilde{\pi}_t(\mathbf{m}) - \pi_t(\mathbf{m}-\boldsymbol{\delta}))\phi(\mathbf{m}+\boldsymbol{\delta})\mathrm{d}\boldsymbol{\delta}\mathrm{d}\mathbf{m}\right| + C\eta \quad (54)$$

$$= \left|\iint_{\Omega_\eta} (\widetilde{\pi}_t(\mathbf{m}) - \pi_t(\mathbf{m}))\phi(\mathbf{m}+\boldsymbol{\delta})\mathrm{Dirac}(\mathrm{d}\boldsymbol{\delta}|T(\mathbf{m};\widetilde{\pi}_t)\Delta t)\mathrm{d}\mathbf{m}\right| + C\eta \quad (55)$$

$$= \left|\iint_{\Omega_\eta} (\widetilde{\pi}_t(\mathbf{m}) - \pi_t(\mathbf{m}))(\phi(\mathbf{m}) + \nabla\phi(\mathbf{m})^\top\boldsymbol{\delta} + \mathcal{O}(\|\boldsymbol{\delta}\|_2^2))\mathrm{Dirac}(\mathrm{d}\boldsymbol{\delta}|T(\mathbf{m};\widetilde{\pi}_t)\Delta t)\mathrm{d}\mathbf{m}\right| + C\eta \quad (56)$$

$$\leqslant \left|\iint_{\Omega_\eta} (\widetilde{\pi}_t(\mathbf{m}) - \pi_t(\mathbf{m}))\phi(\mathbf{m})\mathrm{Dirac}(\mathrm{d}\boldsymbol{\delta}|T(\mathbf{m};\widetilde{\pi}_t)\Delta t)\mathrm{d}\mathbf{m}\right|$$

$$+ \left|\iint_{\Omega_\eta} (\widetilde{\pi}_t(\mathbf{m}) - \pi_t(\mathbf{m}))(\nabla\phi(\mathbf{m})^\top\boldsymbol{\delta} + \mathcal{O}(\|\boldsymbol{\delta}\|_2^2))\mathrm{Dirac}(\mathrm{d}\boldsymbol{\delta}|T(\mathbf{m};\widetilde{\pi}_t)\Delta t)\mathrm{d}\mathbf{m}\right| + C\eta \quad (57)$$

$$\leqslant |\langle \widetilde{\pi}_t - \pi_t, \phi \rangle| + \left|\int_{\Omega_\eta} (\widetilde{\pi}_t(\mathbf{m}) - \pi_t(\mathbf{m}))(\nabla\phi(\mathbf{m})^\top T(\mathbf{m};\widetilde{\pi}_t))\mathrm{d}\mathbf{m}\right| \cdot \Delta t + C\eta + \mathcal{O}(\Delta t^2) \quad (58)$$

$$\leqslant |\langle \widetilde{\pi}_t - \pi_t, \phi \rangle| + \left|\int_{\Omega_\eta} (\widetilde{\pi}_t(\mathbf{m}) - \pi_t(\mathbf{m}))\phi(\mathbf{m})(\nabla(\log\phi(\mathbf{m}))^\top T(\mathbf{m};\widetilde{\pi}_t))\mathrm{d}\mathbf{m}\right| \cdot \Delta t + C\eta + \mathcal{O}(\Delta t^2)$$

$$\leqslant |\langle \widetilde{\pi}_t - \pi_t, \phi \rangle| + \left|\int_{\Omega_\eta} (\widetilde{\pi}_t(\mathbf{m}) - \pi_t(\mathbf{m}))\phi(\mathbf{m})\mathrm{d}\mathbf{m}\right| \cdot \max_{\Omega_\eta}\left|\nabla(\log\phi(\cdot))^\top T(\cdot;\widetilde{\pi}_t)\right|\Delta t + C\eta + \mathcal{O}(\Delta t^2)$$

$$\leqslant (1 + C_t\Delta t)|\langle \widetilde{\pi}_t - \pi_t, \phi \rangle| + C\eta + \mathcal{O}(\Delta t^2). \quad (59)$$

To bound (T1), one need to leverage the localization property of $\nu_t$. Recall that $\nu_t^*$ is the minimizer of

$$\min_\nu \mathcal{L}[\nu * \pi_t], \text{ s.t. } \nu * \pi_t \in \widehat{\mathcal{P}}(M_t^\rho). \tag{60}$$

This implies $\mathcal{L}[\nu_t^* * \pi_t] - \mathcal{L}[\nu_t * \pi_t] \leqslant 0$. Applying the second-order functional expansion at $\nu_t^* * \pi_t$ to the left hand side yields that

$$\left\langle \frac{\delta}{\delta\pi_{t+\Delta t}}\mathcal{L}, (\nu_t^* - \nu_t) * \pi_t \right\rangle \leqslant C_t \left\| \frac{\delta^2}{\delta^2\pi_t}\mathcal{L} \right\|_\infty \cdot \|(\nu_t^* - \nu_t) * \pi_t\|_{L_2}^2 \leqslant C_t\Delta t^2. \tag{61}$$

On the other hand, since $\nu_t$ is the minimizer of

$$\min_\nu \left\langle \frac{\delta}{\delta\pi_t}\mathcal{L}, \nu * \pi_t - \pi_t \right\rangle, \text{ s.t. } \nu * \pi_t \in \widehat{\mathcal{P}}(M_t^\rho) \implies 0 \leqslant \left\langle \frac{\delta}{\delta\pi_t}\mathcal{L}, (\nu_t^* - \nu) * \pi_t \right\rangle, \tag{62}$$

which further implies

$$\left\langle \frac{\delta}{\delta\pi_{t+\Delta t}}\mathcal{L}, (\nu_t^* - \nu_t) * \pi_t \right\rangle \geqslant \left\langle \frac{\delta}{\delta\pi_t}\mathcal{L}, (\nu_t^* - \nu_t) * \pi_t \right\rangle - 2\left\| \frac{\delta^2}{\delta^2\pi_t}\mathcal{L} \right\|_\infty C_t\Delta t^2 \tag{63}$$

$$\geqslant \left| \left\langle L_\varepsilon + \mu_{\pi_t}, (\nu_t^* - \nu_t) * \pi_t \right\rangle \right| - C_t\Delta t^2 \tag{64}$$

$$\geqslant \inf_{M_t^\rho}(L_\varepsilon + \mu_{\pi_t}) \cdot \|(\nu_t^* - \nu_t) * \pi_t\|_{L_1} - C_t\Delta t^2. \tag{65}$$

Therefore, the combination of (61) and (65) implies

$$|\langle (\nu_t - \nu_t^*) * \pi_t, \phi \rangle| \leqslant \|(\nu_t^* - \nu_t) * \pi_t\|_{L_1}\|\phi\|_\infty \leqslant C_t\Delta t^2. \tag{66}$$

For any fixed $\eta > 0$, we combine (59) and (66) and take $\Delta t \to 0$ to derive

$$|\langle \widetilde{\pi}_{t+\Delta t} - \pi_{t+\Delta t}, \phi \rangle| - |\langle \widetilde{\pi}_t - \pi_t, \phi \rangle| \leqslant C_t\Delta t^2 + C_t|\langle \widetilde{\pi}_t - \pi_t, \phi \rangle|\Delta t + C\eta \tag{67}$$

$$\stackrel{\Delta t \to 0}{\Longrightarrow} \quad \mathrm{d}|\langle \widetilde{\pi}_t - \pi_t, \phi \rangle| \leqslant C\eta + C_t|\langle \widetilde{\pi}_t - \pi_t, \phi \rangle|\mathrm{d}t. \tag{68}$$

Finally, we apply the Gröwnwall inequality (Gronwall, 1919) to the function $t \mapsto |\langle \widetilde{\pi}_t - \pi_t, \phi \rangle|$ and we prove that

$$|\langle \widetilde{\pi}_t - \pi_t, \phi \rangle| \leqslant C\eta \left( t + \int_0^t sC_s e^{\int_s^t C_\tau \mathrm{d}\tau}\mathrm{d}s \right) \tag{69}$$

holds for any $t \in [0, 1 - d'/d]$ and any $\eta > 0$. By taking $\eta \to 0$, the right-hand side of (69) tends to 0, which completes the proof.

$\square$

## D.2 REALISING SFPK VIA PARTICLE SIMULATION-BASED APPROXIMATION

In this section, we aim to realize the sampling from $\pi_{1-d'/d}$ by simulating SFPK using an interacting mask particle system. We omit the literature review on some basic concepts, such as infinitesimal generator and the origins of FPK equations, in thermodynamic and stochastic analysis, and we refer interested readers to (Bogachev et al., 2015; Wild et al., 2023) for detailed introductions.

Let $(\mathbf{m}_t)_t$ be a stochastic process, let $\pi_t$ be the particle density function at time $t$, the stochastic analysis theory shows that the time evolution of $\pi_t$ is associated with the adjoint of the infinitesimal generator of the stochastic process. Specifically, suppose $\pi_t$ follows an FPK equation $\partial_t\pi_t = \mathcal{A}^*\pi_t$, where $\mathcal{A}^*$ is a known differential operator. Then, one can instantiate $\pi_t$ using a stochastic process $(\mathbf{m}_t)_t$, such that $\mathrm{law}(\mathbf{m}_t) = \pi_t$ for all $t$, and the infinitesimal generator of the process equals to $\mathcal{A}$, which is indeed the adjoint operator of $\mathcal{A}^*$ (that is $\langle \mathcal{A}[f], g \rangle = \langle f, \mathcal{A}^*[g] \rangle$). Finally, the dynamic of $(\mathbf{m}_t)_t$ can be derived based on its infinitesimal generator $\mathcal{A}$. In intuition, the infinitesimal generator is an operator describing how an arbitrary statistic of the thermodynamic system changes under an infinitesimal time increase.

Following this spirit, we aim to derive the stochastic process associated with SFPK. To this end, we only need to identify the adjoin operator of $\pi_t \mapsto -\nabla \cdot [T(\cdot, \pi_t)\pi_t]$. For any bounded smooth

test function $\phi(\cdot)$, we want to identify $\mathcal{A}$, such that $\langle -\nabla \cdot [T(\cdot, \pi_t)\pi_t], \phi \rangle = \langle \pi_t, \mathcal{A}[\phi] \rangle$. Following standard derivation, it holds that

$$\langle -\nabla \cdot [T(\cdot, \pi_t)\pi_t], \phi \rangle = -\int \nabla \cdot [T(\mathbf{m}, \pi_t)\pi_t(\mathbf{m})]\phi(\mathbf{m})\mathrm{d}\mathbf{m} = \int T(\mathbf{m}, \pi_t)^\top \nabla \phi(\mathbf{m})\pi_t(\mathbf{m})\mathrm{d}\mathbf{m}, \quad (70)$$

which implies $\mathcal{A}[\cdot] = T(\mathbf{m}, \pi_t)^\top \nabla$. According to the stochastic analysis theory (Wild et al., 2023, Equation (54)), the SFPK is associated to the following McKean-Vlasov process:

$$\mathrm{d}\mathbf{m}_t = T(\mathbf{m}_t, \mathrm{law}(\mathbf{m}_t))\mathrm{d}t, \quad (71)$$

where the drift term is dependent on **both the microscopic coordinate $\mathbf{m}_t$ and the macroscopic distribution** $\pi_t$. Fortunately, (Veretennikov, 2006, Section 2) provides a standard approach to simulate solutions to (71) using the empirical distribution over an ensemble of interacting particles. Formally, the authors show that if we take

$$\mathrm{d}\mathbf{m}_t^i = T(\mathbf{m}_t^i, \widehat{\pi}_t^n)\mathrm{d}t, \ i = 1, ..., n, \ \text{with } \widehat{\pi}_t^n \triangleq \frac{1}{n}\sum_{i=1}^n \mathrm{Dirac}(\cdot; \mathbf{m}_t^i), \quad (72)$$

then (Veretennikov, 2006, Theorem 2.2) implies that $\widehat{\pi}_t^n \rightharpoonup \pi_t$ as $n \to \infty$. Therefore, to sample performant mask from $\pi_{1-d'/d}$, we simulate the SFPK dynamic using $n$ interacting mask particles and update the system according to (72) from $t = 0$ to $t = 1 - d'/d$. Finally, we can sample desired masks from the empirical distribution $\widehat{\pi}_{1-d'/d}$.

