# OpenReview forum: "Probabilistic Neural Pruning via Sparsity Evolutionary Fokker-Planck-Kolmogorov Equation"
_ICLR.cc/2025/Conference — ICLR 2025 Spotlight_

### Official Review · Reviewer_Y7sb · 2024-10-19

**Soundness:** 3
**Presentation:** 4
**Contribution:** 4
**Rating:** 8
**Confidence:** 3

**Summary:**

The authors use probabilistic lifting to reformulate gradual pruning as the evolution of the mask distribution. The authors then assume locality and derive the equations governing the sparsity evolution, which the authors refer to as the SFPK equation. Finally, based on SFPK, they propose a new pruning algorithm, which they demonstrate through extensive experiments its SOTA performance on a range of pruning tasks.

**Strengths:**

The paper presents a strong combination of theory and experiments. The paper proposes a theoretically driven algorithm, which achieves the state-of-the-art performance on a range of pruning tasks. The theoretical analyses, especially the relation to the FPK equation, bring new insights for pruning. Overall, the paper has a significant impact to the field.

**Weaknesses:**

Major points:

1.	The authors does not emphasize the critical role of using mini batches in algorithm 1. Mini-batches and stochastic noises are crucial for algorithm 1. If using full batches, then T should be the same for all m_i from the begining, resulting a collapse of the distribution.

2.	The performances of SNIP and SynFlow in table 1 seem unreasonably low. I suspect there may be some mistakes in the experiments. Specifically, both methods prune the network at initialization prior to training. I wonder if the authors mistakenly apply those pruning algorithms at the end of training. This could explain the discrepancies in performance, and I recommend the authors review the experimental setup for these pruning algorithms to ensure correctness.


Minor points:

1.	It may be good to define the distributional transition T when it is first introduced at the end of section 3.

2.	I don’t find a definition of the mask polarizer $P_\epsilon$ in the main text. It should be defined.

3.	The authors claim their algorithm is robust against different $\lambda$, but this should be expected as $\lambda$ is introduced for a technical reason to convexify the optimization problem. From an algorithm perspective, it may not be necessary to have $\lambda$ at all. It would also be useful to include results in Fig. 3 with $\lambda=0$.

**Questions:**

Please refer to the weakness.

---

> ### Author Response · Authors · 2024-11-23
> **Response to Reviewer Y7sb**
>
> We sincerely appreciate Reviewer Y7sb's time, effort, insightful feedback, and constructive suggestions, which are valuable in helping us refine our work. Below, we address the concerns raised and provide additional discussions and justifications to clarify our contribution.
>
> ----
> **W1. Using mini-batch gradient in SFPK.**
>
> **Response to W1.** We thanks the Reviewer-Y7sb for his valuable comment and we will add the following discussions into the Appendix A.1.
>
> "Specifically, we would like to emphasize that the mini-batch gradients involved in Algorithm 2 is crucial for mask particle simulation. In theory, the SFPK dynamic should be initialized at $\pi_0^*$, the optimal mask distribution within $M_0^{\rho}$. Since $\rho > 0$, $\pi_0^*$ is closed to, but not identical to, the delta distribution centered on the fully dense mask. In practice, we approximate $\pi_0^*$ by running SFPK with stochastic mini-batch gradient from the delta distribution at the fully dense model. During the SFPK particle simulation process, the stochasticity of the mini-batch gradients enhances robustness and stability in sampling, while also encouraging the exploration of diverse mask particles."
>
> -----
> **W2. Performance of SNIP and SynFlow in Table 1.**
>
> **Response to W2.** We have thoroughly checked our SNIP and SynFlow codes and confirm they match the official release with recommended hyperparameters (see Section 13.1 of [1]). Our results shows that SNIP and SynFlow fails to achieve satisfactory results in the one-shot pruning without retraining scenario.
>
> We confirm that SynFlow does not consistently work well, likely due to:
>
> 1. As a pruning-at-initialization (PAI) method, SynFlow is less effective for post-training pruning (PTP) as its data-agnostic SynFlow score is less informative than data-dependent pruning scores when the data and a well trained weight are accessible.
> 2. As noted in Section 12 in [1], the SynFlow score can be overly large and numerically unstable. While rescaling weights can mitigate this in PAI, it is not applicable to PTP and may harm the pretrained weights. Hence, SynFlow fails to generalize to new architectures (e.g., ResNet-18/20, WRN-19/20, VGG-16/16bn) and weights (e.g., randomly initialized / pretrained) with the recommended hyperparameter.
>
> We ran PAI experiments on VGG16 CIFAR-100 at 90% sparsity using our code and the official SynFlow code. Our code achieved 56.41% accuracy, while the official code achieved 56.34%, showing that our SynFlow code is correct. This can be verified by running the following lines in the official SynFlow Github repo:
>
> ```bash
> python main.py --experiment singleshot --model vgg16 --dataset cifar100 --model-class lottery --optimizer momentum --train-batch-size 128 --pre-epochs 0 --post-epochs 160 --lr 0.1 --lr-drops 60 120 --lr-drop-rate 0.1 --weight-decay 0.0001 --pruner synflow --compression 1.0 --prune-epochs 100 --mask-scope global
> ```
>
> We also confirm that SNIP does not consistently perform well, likely because the Taylor expansion score it uses becomes less effective when a significant number of parameters are removed simultaneously. Furthermore, we conjecture that its limited performance stems from SNIP's greedy pruning strategy, which prevents it from regrowing prematurely pruned weights, potentially leading to suboptimal results.
>
>
> [1]. Hidenori Tanaka, Daniel Kunin, et al. Pruning neural networks
> without any data by iteratively conserving synaptic flow. (NeurlPS 2020)
>
> -----
> **W3. Definition of distributional transition $\mathcal{T}[\cdot]$.**
>
> **Response to W3.** In particular, the closed-form expression of the distributional transition $\mathcal{T}[\pi_t]$ is given by $- \nabla \cdot[T(\cdot; \pi_t) \pi_t]$ as proven in Proposition 3. Therefore, it is difficult to introduce the formal definition of $\mathcal{T}[\cdot]$ until Proposition 3.
>
> To enhance clarity, we include a footnote after Equation (2) stating, "The formal expression of the distributional transition $\partial_t\pi_t =\mathcal{T}[\pi_t]$ will be introduced in Proposition 3."
>
>
>
> -----
> **W4. Definition of polarizer.**
>
> **Response to W4.** We add the formulation of mask polarizer in Section 4.1. And we refer the readers to the Appendix A.3 in the original paper of PSO [1.] for more discussions and implementation details of the mask polarizer.
>
> [1.]. Zhanfeng Mo, Haosen Shi, and Sinno Jialin Pan. Pruning via sparsity-indexed ODE: a continuous sparsity viewpoint. (ICML 2023)
>
> -----
> **W5. $\lambda=0$ result in Figure 3.**
>
> **Response to W5.** We have added the $\lambda=0$ result in Figure 3 (left). In practice, SFPK with $\lambda=0$ still achieves decent performance compared to the iterative magnitude pruner. However, increasing $\lambda$ beyond 0 leads to further performance improvements.

---

> > ### Comment · Reviewer_Y7sb · 2024-11-24
> >
> > The authors have addressed most of my concerns, except for the following:
> >
> > Regarding the performance of SNIP and SynFlow, these algorithms are specifically designed for pruning at initialization rather than post-training. Therefore, comparing their post-training one-shot pruning performance is not a fair evaluation. A more appropriate experiment would involve pruning at initialization, followed by training the network to completion, and then reporting the final performance.

---

> ### Author Response · Authors · 2024-11-25
> **Valid pruning at initialization experiment has been added.**
>
> Dear Reviewer Y7sb,
>
> Thanks for your reply.  We have included valid comparison on CIFAR-100, WRN-20 under the pruning-at-initialization setting in the revised manuscript.
>
> As mentioned in both the **Summary of rebuttal revision, Line 8** and **our Response to Reviewer 3E6s-Q2, Part 2**, we compare our SFPK against SNIP, SynFlow, GraSP, and Mag under the pruning-at-initialization (PaI) setting. Specifically, we randomly intialize a WRN-20, prune it to 95% sparsity, and retrain it for  epochs to convergence. The average test accuracy among of final  epochs is reported based on 3 trials.
>
> As shown in Appendix C, Table 9, our SFPK illustrates better performance than other baselines. Furthermore, the performance of SFPK can be evidently improved when a well-trained checkpoint is provided. This evidence helps clarify the significance of the 'rational' constraint.
>
> Due to the limited time and computation resource, the PaI experiment on ImageNet is still pending (i.e. the one-shot prune w/ fine-tuned version of Table 1). We would add the associated results to the final version as long as the experiments are completed.

---

> > ### Comment · Reviewer_Y7sb · 2024-11-25
> >
> > Thank you to the authors for the clarification. I apologize for overlooking the newly added Table 9, which effectively addresses my concerns. While I will maintain my current score, I want to emphasize that I consider this to be a strong paper.

---

> > > ### Author Response · Authors · 2024-11-25
> > >
> > > Dear Reviewer Y7sb,
> > >
> > > We sincerely appreciate your time, effort, insightful feedback, and constructive suggestions, which have been invaluable in helping us improve our work.
> > >
> > > Best,
> > >
> > > Authors

---

### Official Review · Reviewer_3JmV · 2024-10-28

**Soundness:** 3
**Presentation:** 3
**Contribution:** 3
**Rating:** 8
**Confidence:** 5

**Summary:**

Summary

- In this paper, the authors reformulate pruning as an expected loss minimization problem within the mask distribution space and introduce the SFPK to represent distributional transitions. Leveraging the proposed SFPK, they develop the SFPK-pruner, a particle simulation-based probabilistic pruning method.

**Strengths:**

Strengths

- The paper is well-written and easy to follow.
- Using the probabilistic lifting and convexification technique from [1] to develop a probabilistic soft neural pruning approach is a fresh and intriguing method.
- The theoretical analysis and approximations supporting the SFPK-guided pruning proposal are reasonable and add an interesting perspective.
- Overall, I am satisfied with the experimental scale, results, and novelty of the method. Although the requirement to create $ n $ mask particles presents practical challenges for applying this method to large models, such as large language models, I still believe this is a good paper that falls within the acceptance boundary.

References

[1] Veit David Wild, Sahra Ghalebikesabi, Dino Sejdinovic, and Jeremias Knoblauch. A rigorous link between deep ensembles and (variational) bayesian methods. In Thirty-seventh Conference on Neural Information Processing Systems, 2023.

**Weaknesses:**

Weaknesses

- Additional analysis on the computational overhead as $ n $ and $ K $ increase would be beneficial. While these values can be adjusted, a larger $ n $ could lead to a linearly increased memory overhead, and a larger $ K $ could introduce extra computational burden compared to other methods.

- A comparison with similar methods, such as iterative magnitude pruning methods, which also gradually adjust sparsity, would provide useful context.

- It would be helpful to assess the performance of each of the $ n $ particles individually and check if they display functional diversity. Verifying whether the method adequately covers the posterior distribution of the mask or simply converges towards one or a few local minima with good performance would more clearly demonstrate the effectiveness of the proposed method.

**Questions:**

See Weaknesses section. If the responses to my questions are satisfactory, I am open to increasing my rating.

---

> ### Author Response · Authors · 2024-11-23
> **Response to Reviewer 3JmV**
>
> We sincerely appreciate Reviewer 3JmV's time, effort, insightful feedback, and constructive suggestions, which are valuable in helping us refine our work. Below, we address the concerns raised and provide additional discussions and justifications to clarify our contribution.
>
> -----
> **W1. Computational overhead w.r.t $n$ and $K$.**
>
> **Response to W1.** A detailed complexity analysis of SFPK is included in Appendix A.1.
>
> In particular, the cost of applying SFPK pruner (with $n$ discretization steps and $K$ mask particles) equals to training the model for $n\times K$ mini-batches plus a marginal $O(nKd)$ inner-product computation.
>
> -----
> **W2. Comparison with iterative magnitude pruning.**
>
> **Response to W2.** In Table 2b and Table 2c in Section 5.2, we compared SFPK against two iterative magnitude pruning baselines, i.e. the Mag (iterative global magnitude pruning) and GMP + LS (iterative magnitude pruning with improved sparsity configuration) [1]. The experiment shows that SFPK outperforms iterative magnitude pruning variants on both MobileNet-V1 and ResNet-50, under ImageNet-1K.
>
> [1]. Trevor Gale, Erich Elsen, et al. The state of sparsity in deep neural networks, 2019.
>
> -----
> **W3. Diversity of SFPK particles.**
>
> **Response to W3.** We conduct additional experiment to study the diversity of the simulated particle distribution of SFPK. Specifically, at each intermediate step of SFPK, we track the mean and standard deviation of the loss function (the energy) and the average sparsity deviation of each mask particle. The average relative mask deviation of the $i$-mask is computed by  $1/(n-1)\sum_j\|\mathbf{m}^i_t - \mathbf{m}^j_t\|_2^2/ \|\mathbf{m}^i_t\|_2^2$. We visualize the mean and standard deviation (across $10$ mask particles) of both the loss value and the relative mask deviation.
>
> As shown in Figure 5, Appendix C, as sparsity increases, the energy of each mask increases without drastic explosion. Meanwhile, the diversity of the masks progressively grows. This suggests that the SFPK mask distribution does not collapse and it generates diverse yet effective mask particles as sparsity decreases, allowing us to sample performant masks at the desired sparsity level.

---

> ### Comment · Reviewer_3JmV · 2024-11-23
>
> Thank the authors for the additional experiments and discussions. I like this paper and appreciate the effort put into it, so I will increase my score from 6 to 8.

---

> > ### Author Response · Authors · 2024-11-24
> >
> > We would like to once again express our heartfelt gratitude to you for dedicating your time and effort to reviewing our rebuttal and paper.
> >
> > We remain open and willing to address any further questions or concerns you may have.

---

### Official Review · Reviewer_TsAq · 2024-10-30

**Soundness:** 3
**Presentation:** 3
**Contribution:** 3
**Rating:** 6
**Confidence:** 4

**Summary:**

This paper introduces a method to identify a sparse network by incrementally selecting the mask that results in the smallest performance reduction as sparsity increases. Deriving from  thermodynamics, the authors begin with a population of dense masks and progressively evolve a distribution of sparse masks using the Fokker-Planck-Kolmogorov equation. Their approach offers a way to transition to a higher sparsity mask distribution by increasing the sparsity by a small amount in every step. Additionally, the authors propose a pruning scheme based on this transition, which iteratively refines a population of sparse masks to achieve an effective sparse network.

**Strengths:**

1. The proposed algorithm optimizes a population of sparse mask in an evolutionary manner while increasing sparsity to find a performant sparse network. This is a novel which seems to dynamical learn mask structure similar to dynamic sparse training method albeit in an evolutionary manner.
2. The authors also experimentally validate the proposed pruning method.

**Weaknesses:**

1. The authors provide a comprehensive experimental evaluation of the proposed method. However, the performance across tasks is similar to dynamic sparse training methods like RiGL and only slightly outperforms it in some cases (as in Table 8). Given this, it would be important to know the computational comparisons of DST methods versus the SFPK pruner. On ImageNet the authors mention that the SFPK pruner adds an overhead equivalent to 3 training epochs, would this mean it is marginally more expensive than DST?.
2. In order to optimize the mask, the sparsity function chosen by the authors is the polarizer. Is there a reason for choosing this specific operator, how would it compare with other operators such as soft thresholding or even a sigmoid.
3. Results in Table 1 could be a bit misleading since the pruning at initialization methods like Snip and Grasp are not designed to minimize the loss upon sparsification and hence cannot be compared without any retraining after pruning.

**Questions:**

See above.

---

> ### Author Response · Authors · 2024-11-23
> **Response to Reviewer TsAq**
>
> We sincerely appreciate Reviewer TsAq time, effort, insightful feedback, and constructive suggestions, which are valuable in helping us refine our work. Below, we address the concerns raised and provide additional discussions and justifications to clarify our contribution.
>
> -----
> **W1. Computation cost vs DST methods.**
>
> **Response to W1.** As shown in Appendix C, we compare the training and inference FLOPs of SFPK with 3 dynamic sparse training (DST) methods, i.e. RigL [1] (Table 14), SWAMP [2] (Table 12), and MEST [3] (Table 16). We reported the training FLOPs in units of $1e18$ and inference FLOPs in units $1e9$.
>
> Specifically, Table 14 indicates that SFPK is less expensive than RigL in terms of training FLOPs.
>
> [1]. Utku Evci, Trevor Gale, et al. Rigging the lottery: Making all tickets winners. (ICML 2020)
>
> [2]. Moonseok Choi, Hyungi Lee, et al. Sparse weight averaging with multiple particles for iterative magnitude pruning. (ICLR 2024)
>
> [3]. Geng Yuan, Xiaolong Ma, et al. Mest: Accurate and fast memory-economic sparse training framework on the edge. (NeurlPS 2021)
>
>
> -----
> **W2. Reason for using polarizer.**
>
> **Response to W2.** To analyze mask evolution under an infinitesimal $dt$-sparsity increment, we generalize discrete masks to continuous-valued and nearly-sparse soft masks. Accordingly, the hard sparsity measure $1 - \|m\|_0 / d$ is extended to a soft sparsity measure $1 - \|m\|_2 / d$.
>
> To study the sparsity evolution of masks, an ideal polarization operator $P(m)$ is expected to preserve the sparsity of soft masks, satisfying the condition $1 - \|m\|_2 / d = 1 - \|P(m)\|_2 / d$. In this case, the soft sparsity of the nearly-sparse mask $P(m)$ can be evaluated directly by computing the $\ell_2$ norm of the soft, unpolarized mask $m$.
>
> In contrast, standard softmax or sigmoid operators do "sparsify" the mask, but they fail to preserve the soft sparsity of the resulting "sparsified masks." This issue can be resolved through renormalization, i.e. setting
> $P(m) = \text{softmax}(m) \cdot \|m\|_2 / \|\text{softmax}(m)\|_2.$
>
> Generally, designing better mask polarization operators for neural pruning is an orthogonal research direction to this paper, and we will postpone this to future work.
>
>
> -----
> **W3. Comparisons in Table 1.**
>
> **Response to W3.** In the post-training scenario (e.g. pruning a pretrained dense model), SFPK shares the same objective with Mag, SNIP, and WF: all of them are primarily designed to minimize the pruning-induced accuracy drop (i.e. pruning error).
>
> We will consider removing the results of SynFlow from Table 1, as it was not designed to minimize the loss upon sparsification.
>
> To minimize loss upon sparsification, our SFPK use a particle simulation approach, while Mag, SNIP, and WF minimize a Taylor-expansion-based pruning error bound, i.e. $L(\theta\odot m) - L(\theta)\approx \nabla L(\theta)^\top \delta\theta+1/2\ \delta\theta^\top \nabla^2L(\theta)\delta\theta$, where $\delta\theta\triangleq (\theta\odot m) - \theta$ is the mask perturbation.  Then, Mag aims to minimize $\|\delta\theta\|$ and SNIP aims to minimize $\|\nabla L(\theta)^\top\delta\theta\|$, both neglecting the second-order term. In contrast, WF aims to minimize $\|\delta\theta^\top  \nabla^2L(\theta)\delta\theta\|$, neglecting the first order term.
>
> Empirically, our Table 1 shows SFPK is more beneficial in reducing pruning error. It validates that our SFPK theory provides a better approximation of the optimal masks than other baseline score-based methods (e.g., Mag, SNIP, and WF).

---

> > ### Comment · Area_Chair_QJhj · 2024-11-25
> > **The author-reviewer discussion period is ending soon.**
> >
> > Dear reviewer,
> >
> > Please engage in the discussion as soon as possible. Specifically, please acknowledge that you have thoroughly reviewed the authors' rebuttal and indicate whether your concerns have been adequately addressed. Your input during this critical phase is essential—not only for the authors but also for your fellow reviewers and the Area Chair—to ensure a fair evaluation.
> > Best wishes,
> > AC

---

> > ### Comment · Reviewer_TsAq · 2024-11-25
> > **Reponse to author rebuttal**
> >
> > Thank you for providing a comprehensive response and apologies for the delay in my reply.
> >
> > 1. Table 14 and 16 provide a comprehensive analysis of the computational expense of SFPK.
> > 2. Thanks for the clarification regarding the polarizer.
> > 3. Yes, I believe adding the explanation that ‘SFPK is more beneficial to reduce pruning error’ will provide important context to these results. However, Snip and Synflow are still do not help to reduce the pruning error as they are enforced at initialization.
> >
> > I am happy with the author response and will keep my score.

---

> > > ### Author Response · Authors · 2024-11-25
> > >
> > > Dear Reviewer TsAq,
> > >
> > > We sincerely appreciate your time, effort, insightful feedback, and constructive suggestions, which have been invaluable in helping us improve our work.
> > >
> > > Best,
> > >
> > > Authors

---

### Official Review · Reviewer_3E6s · 2024-11-04

**Soundness:** 3
**Presentation:** 3
**Contribution:** 3
**Rating:** 8
**Confidence:** 3

**Summary:**

This work formulates the neural network pruning procedure as the evolution of sparse mask distributions in response to increasing levels of sparsity, employing the Fokker-Planck-Kolmogorov equation. Specifically, given a rational dense solution, it provides a theoretically well-founded definition of the distributional transition for the sparse mask, indicating which parameters should be pruned. Experimental results are provided for both one-shot and iterative schemes, as well as for structured and unstructured sparsity, covering convolutional and vision transformer architectures.

**Strengths:**

S1. In neural network pruning, a line of research focuses on iterative methods that start with a dense network with zero sparsity and gradually increase the sparsity level. A well-known example is the Iterative Magnitude Pruning (IMP) algorithm, which, while computationally expensive, is highly effective at producing high-performance sparse networks. This work shares a similar spirit (especially in the prune-and-retrain setup), deriving the sparse mask for each new sparsity level from the previous one, gradually updating the mask as sparsity increases. However, rather than treating the optimal mask at each level as a fixed point estimate, this work consider a distribution over masks, adding a compelling dimension to the method.

S2. As the authors noted, the most closely related work is the ODE framework proposed by Mo et al. (2023), which evolves sparse masks over time by treating the sparsity level as the timestep. This work explicitly acknowledges this connection in the main text and provides a clear discussion in the Introduction that highlights the contributions unique to them. It not only gives proper credit to prior work but also helps readers understand the progression of ideas.

S3. The proposed SFPK methodology is well-motivated, and the flow of Sections 1-4 effectively presents it. Overall, the experiments are well-executed, featuring a comprehensive comparison to various baselines. While there is room for improvement, as noted in the Weaknesses section, the core results are largely well presented.

**Weaknesses:**

Since I appreciate the underlying concepts of the proposed SPFK methodology, I will primarily concentrate on addressing the experimental concerns here.

W1. The tables do not include error bars, making it unclear how statistically significant the current results are. How many trials were conducted for the values presented? From the caption of Figure 2, is it three trials for all experiments? If space is a concern, the main text could provide only the averaged numbers, while a full table with standard deviations could be included in the appendix.

W2. In the _One-shot Pruning_ experiment in Section 5.2, it is quite intriguing that the SPFK method achieves such performance after one-shot pruning on a given pre-trained dense weight without requiring further training for recovery. However, it is important to note that all three methods—WF, PSO, and SFPK—incur significantly higher costs compared to Mag, SNIP, and SynFlow for obtaining valid results. Including a cost comparison for each method would be beneficial. At the very least, the costs associated with SFPK compared to WF and PSO should be presented (especially since Mag, SNIP, and SynFlow would not even be considered when targeting high sparsity).

W3. In the _One-shot Pruning_ experiment in Section 5.2, Table 1a presents results for the convolutional architecture, while Table 2a displays results for the vision transformer architecture. However, Table 1a only includes results _without_ further training and _unstructured_ sparsity, whereas Table 2a only includes results _with_ further training and _structured_ sparsity. Is there a specific reason for this distinction? From the reader's perspective, it would be beneficial to see comprehensive results for both convolutional and vision transformer architectures for each method—{Mag, SNIP, SynFlow, WF, PSO, SFPK}, particularly for {WF, PSO, SFPK} as they are the main competitors—across all scenarios: with and without further training, and structured and unstructured sparsity.

W4. The SFPK methodology fundamentally assumes "rational" pre-trained weights, denoted as $\theta^\ast$. I noticed that the paragraph _"Constraints of SPFK-pruner"_ in Appendix A.1 discusses this topic, stating, _"We recommend applying SFPK to sufficiently trained "rational" models (e.g., trained to convergence)."_ However, there is currently no experimental evidence supporting the practical significance of this recommendation. It is particularly important in the prune-and-retrain setup, as the extent of retraining to re-obtain the "rational" $\theta^\ast$ can consume a substantial portion of the total training budget. I initially expected the Appendix B to address this (after reading lines 448-449), but it still fixes the retraining budget for SFPK at 100 epochs. Currently, there is no discussion on how "rational" pre-trained weights need to be for SFPK to function effectively. Including empirical results on this matter would undoubtedly strengthen the paper.

W5. As of 2024, 90% sparsity for ResNet50 on ImageNet is not considered particularly high (especially when all parameters, including the final linear weights, are allowed to be pruned). To convincingly demonstrate the effectiveness of the proposed method, comparative results should be presented in a higher sparsity regime of 95% or more. Additionally, the comparisons primarily involve relatively old baseline methodologies, with the most recent baseline in Table 2c dating back to 2020; it raises the question of whether it is appropriate to claim that _"our SFPK-pruner demonstrates decent performance compared to SOTA baselines."_ For instance, the method called Spartan (Tai et al., NeurIPS 2022) reports an accuracy of 76.17% for 90% sparsity in a 100-epoch setup. Although the pruning community has shifted its focus away from the ResNet50 on the ImageNet setup, and I am not sure whether recent SOTA results are being reported, it remains important to include comparisons with the latest baselines available in this paper.

P.S. As I continue writing, I recognize that providing such extensive experimental results could be burdensome, especially during the rebuttal period. I want to clarify that I do not expect all results to be presented within this limited timeframe (particularly, feel free to skip W3). Please share any results you can within your available time and resources, along with a brief explanation of how you plan to address the mentioned concerns in the future if they are not currently available.

**Questions:**

Q1. In Lines 428-430, it states that _"the SFPK-pruner with a batch size of $256$ for $n \times K = 1500$ steps is equivalent to that of only ONE epoch of retraining on ImageNet."_ While I understand that running $n = 10$ particles sequentially for $K = 150$ steps (without parallelization) results in a batch size of $256$ over $n \times K = 1500$ steps, it seems that $256 \times 1500 = 384000$ samples would correspond to approximately 0.3 epochs of ImageNet (given that the total number of training samples in ImageNet is 1.28 million), rather than one epoch. Could you clarify this point?

Q2. Regarding W4, can SPFK operate with randomly initialized weights? While this clearly disregards the "rational" constraint, it is noteworthy that single-shot approaches like SNIP, GraSP, and SynFlow originally claim to find effective sparse masks using only randomly initialized weights (i.e., pruning-at-initialization). I am curious whether SPFK can outperform these methods in identifying a better sparse mask in a pruning-at-initialization manner. Ideally, the relationship would be random mask < SNIP, GraSP, SynFlow < SPFK. Alternatively, demonstrating that SPFK does not outperform SNIP, GraSP, and SynFlow in the pruning-at-initialization setup could also help clarify the significance of the "rational" constraint.

Q3. The core of the SFPK pipeline, particularly when compared to Mo et al. (2023), revolves around simulating $n>1$ particles. The paragraph _"Ablation on the Mask Particle Simulation Scheme"_ in Section 5.3 illustrates that the proposed SFPK significantly relies on both $n$ and $K$. Notably, the difference between using $n=1$ and $n>1$ is considerable (in Figure 3), highlighting the importance of leveraging multiple particles for the effectiveness of the SFPK methodology. I'm interested in the diversity of the particles in this context. Intuitively, at low sparsity, there are likely many mask candidates that yield effective sparse solutions. However, at high sparsity, could the particles collapse? If that’s the case, it may be advantageous to reduce the number of simulated particles as sparsity increases, potentially enhancing computational efficiency.

Q4. Are all trainable parameters, including convolutional weights, batch normalization scales and biases, and dense weights and biases, subject to pruning? Some studies on pruning convolutional neural networks focus solely on convolutional weights, e.g., the SWAMP paper (Choi et al. 2024) notes, _"we exclusively perform weight pruning on the convolutional layer, leaving the batch normalization and fully-connected layers unpruned."_ This can lead to discrepancies in reported numbers across studies, particularly in the ResNet50 on ImageNet setting, where the final linear layer accounts for about 8% of the total 25,557,032 parameters (i.e., 2,048,000/25,557,032). Could you provide further details on this? It is important to note that allowing the pruning of the final linear weights often leads to significantly better performance at the same level of sparsity (when we compute the sparsity as the number of unpruned parameters divided by the number of all parameters, regardless of such constraints).

Specifically, the SWAMP results for ImageNet presented in Table 6 indicate 88.9% sparsity, with approximately 2.8 million unpruned parameters, of which 2.0 million are linear weights. When calculating sparsity based solely on prunable parameters  for SWAMP (i.e., convolutional weights), we arrive at approximately 96.9% sparsity. Recalculating SWAMP's sparsity as the number of pruned prunable parameters divided by the total number of prunable parameters yields the following adjustments (see the code block below for details): 80.0% becomes 87.2%, 86.0% becomes 93.7%, and 88.9% becomes 96.9%. While this value may not perfectly align with the sparsity of SFPK, it is noteworthy that SWAMP achieves 75.69% accuracy at 87.2% sparsity, whereas SFPK achieves 75.65% accuracy at 87.4% sparsity, indicating a close alignment in results. Therefore, it is important to assess whether fair comparisons have been adequately conducted for not only SWAMP but also other methods.

```
ResNet50 for ImageNet.
- 25557032 : the total number of parameters in ResNet50 for ImageNet.
- 23454912 : the number of convolutional parameters.
- 2102120 : the number of remaining parameters.

If there are 5111406 (= 3009286 + 2102120) unpruned parameters,
- 1 - 5111406 / 25557032 = 80.0% sparsity.
- 1 - 3009286 / 23454912 = 87.2% sparsity, if we consider only the prunable parameters.
```

Q99. It's just a very minor detail, but I'm curious—why is it referred to as SFPK instead of SEFPK?

---

> ### Author Response · Authors · 2024-11-23
> **Response to Reviewer 3E6s (Part 1, responses to W1, W2, W3, and W4)**
>
> We sincerely appreciate Reviewer 3E6s's time, effort, insightful feedback, and constructive suggestions, which are valuable in helping us refine our work. Below, we address the concerns raised and provide additional discussions and justifications to clarify our contribution.
>
> ------
> **W1. Error bars in tables.**
>
> **Response to W1.** All the results presented in Table 1 and Table 2 are computed based on three trials. We postpone the full results in Table 5 and Table 6 of Appendix C.
>
> ------
> **W2. Cost comparison in oneshot pruning.**
>
> **Response to W2.** In Table 7, Appendix B, we report the actual and theoretical runtime complexities of WF, PSO, and our SFPK-pruner for the one-shot pruning experiment on ImageNet-1K with ResNet-50 and MobileNet-V1 at $80\%$ sparsity. Our results show that, in practice, SFPK outperforms WF while maintaining a comparable computational cost. As expected, the computational cost of SFPK is approximately $10$ times that of PSO, as both methods use $700$ discretization steps, while SFPK additionally simulates $10$ particles.
>
> We also compare the theoretical complexity of our SFPK-pruner with that of PSO and WF. Let $m$ denote the data batch size, $d$ the model size, $K$ the discretization step for both PSO and SFPK-pruner, and $n$ the number of particles in the SFPK-pruner. The theoretical runtime complexity of the SFPK-pruner is $O(nK(d+m))$, the theoretical runtime complexity of PSO is $O(K(d+m))$, and the theoretical runtime complexity of WF is $O(md^2)$. As we will show, for large models where $d$ is extremely large, WF becomes significantly expensive due to the need to compute the Hessian matrix. As expected, the computational cost of SFPK is approximately $n$ times that of PSO, as both methods use $K$ discretization steps, while SFPK additionally simulates $n$ particles.
>
> [1]. Sidak Pal Singh and Dan Alistarh. Woodfisher: Efficient second-order approximation for neural network compression. (NeurlPS 2020)
>
> ----
> **W3. Additional oneshot pruning results.**
>
> **Response to W3.** Following the same configuration as described in Section 5.2, we perform one-shot unstructured and structured pruning experiments on the MobileNet-V1, ResNet-50, and DeiT-T architecture to provide a more comprehensive evaluation. As shown in Appendix B, Table 6, our SFPK exhibits competitive performance against other baselines across various scenarios.
>
> Due to limited computational resources, we are unable to complete the one-shot pruning with tuning results during the rebuttal period. However, we will make every effort to include it in the final version. Nonetheless, the gradual pruning results shown in Table 2 have partially validated the effectiveness of our SFPK in the prune-and-retrain scenario.
>
>
> -----
> **W4. How rational $\theta^\*$ needs to be in SPFK?**
>
> **Response to W4.** To study how the rationality of $\theta^*$ affects the effectiveness of SFPK, we conduct an ablation study on the initialization of SFPK on WRN-32x4 CIFAR-100. Specifically, we first pretrain a dense WRN-32x4 from scratch for 300 epochs to convergence, and we save the intermediate model checkpoints $\theta_t^*$ at training epoch  $t\in\{0, 50, 100, 150, 200, 300\}$. Then, we prune each $\theta_t^*$ to $90\%$ sparsity and retrain it for 100 epochs. We report the training accuracy of each setting based on 3 trials.
>
> As shown in Appendix C, Table 8, the peformance of SFPK increases as the pretraining epochs increases. Specifically, the performance of SFPK retrains at a decent level even when the model is not pretrained to convergence. This empirical evidence suggests the robustness of SFPK w.r.t to the rationality of the model weight.

---

> ### Author Response · Authors · 2024-11-23
> **Response to Reviewer 3E6s (Part 2, responses to W5, Q1, Q2, and Q3)**
>
> **W5. Pruning ResNet-50 on ImageNet.**
>
> **Response to W5.** Following the setup in Table 2c, we prune ResNet-50 to the 95% sparsity level using SFPK with the same hyperparameters. Due to the limited computation resource, the reported accuracy is based on one trial, and we will add the results of $2$ extra trials into the final version. As shown in Table 10, Appendix C, our SFPK outperforms WF, STR, GMP, DNW, and RIGL+ERK.
>
> In particular, the experiments in this paper are designed to validate the effectiveness of the proposed SFPK theory in approximating the sparsity evolution of the optimal mask distribution in neural pruning. Thus, we mainly focused on the pruning settings, where the mask and weights are not jointly optimized as in sparse training methods, e.g. `RiGL`, `Spartan`. We have added a reference to `Spartan` in the related work section. Extending SFPK to the sparse training scenario, which requires considering the joint evolution of optimal masks and weights, is deferred to future work.
>
> We have also compared our SFPK-pruner against L1-spred [2] in Table 13 and Figure 6, Appendix C, a SOTA L1 pruning method proposed in 2023. For comprehensiveness, we further include the results of CrAM [3], another pruning method proposed in 2023, into our Table 2c.
>
> To improve clarity, we revise the claim from "*our SFPK-pruner demonstrates decent performance compared to SOTA baselines*" to "*our SFPK-pruner demonstrates decent performance compared to various competitive pruning baselines*".
>
>
> [2]. Liu Ziyin and Zihao Wang. spred: Solving l1 penalty with SGD. (ICML 2023).
>
> [3] Peste, A., et al. CrAM: A Compression-Aware Minimizer. (ICLR 2023).
>
>
> -----
> **Q1. Typo in SFPK overhead.**
>
> **Response to Q1.** We apologize for this typo. The phrase "*... that of only ONE epoch ...*" should be "*... that of only 0.3 epoch ...,*" as stated in the complexity analysis section in Appendix A.1.
>
>
> ----
> **Q2. SPFK with random initialization.**
>
> **Response to Q2.** We compare our SFPK against SNIP, SynFlow, GraSP, and Mag under the pruning-at-initialization setting. Specifically, we randomly intialize a WRN-20, prune it to 95\% sparsity, and retrain it for $100$ epochs to convergence. The average test accuracy among of final $5$ epochs is reported based on 3 trials.
>
> As shown in Appendix C, Table 9, our SFPK illustrates better performance than other baselines. However, the performance of SFPK can be evidently improved when a well-trained checkpoint is provided. This evidence helps clarify the significance of the 'rational' constraint.
>
>
> -----
> **Q3. Diversity of SFPK particles.**
>
> **Response to Q3.** We conduct additional experiment to study the diversity of the simulated particle distribution of SFPK. Specifically, at each intermediate step of SFPK, we track the mean and standard deviation of the loss function (the energy) and the average sparsity deviation of each mask particle. The average relative mask deviation of the $i$-mask is computed by $1/(n-1)\sum_j\|\mathbf{m}^i_t - \mathbf{m}^j_t\|_2^2/ \|\mathbf{m}^i_t\|_2^2$. We visualize the mean and standard deviation (across $10$ mask particles) of both the loss value and the relative mask deviation.
>
> As shown in Figure 5, Appendix C, as sparsity increases, the energy of each mask increases without drastic explosion. Meanwhile, the diversity of the masks progressively grows. This suggests that the SFPK mask distribution does not collapse and it generates diverse yet effective mask particles as sparsity decreases, allowing us to sample performant masks at the desired sparsity level.

---

> ### Author Response · Authors · 2024-11-23
> **Response to Reviewer 3E6s (Part 1, responses to Q4 and Q5)**
>
> **Q4. Pruning scope and sparsity calculation.**
>
> **Response to Q4.** We follow the pruning scope and sparsity calculation as in Section S7.1 of WoodFisher [4] (also in DPF [5] and STR [6]): only the weights in fully-connected and convolutional layers are pruned, and none of the batch-norm and bias parameters are pruned if presented. In this paper, we consistently evaluate the sparsity based on '1 - (\# remaining parameters) / (\# total parameters in convolutional and linear layers)' across different pruning and sparse training methods.
>
> As our denominator (which equals $25502912$) is larger than '\# convolutional parameters' (which equals $23454912$), the sparsity metric adopted in our paper is generally higher and more restrictive than the metric '1 - (\# remaining parameters) / (\# convolutional parameters)' when given the same sparsity level.
>
> Since we consistently use the sparsity metric '1 - (# remaining parameters) / (# total parameters in convolutional and linear layers)', "avoiding pruning linear layers" in SWAMP [7] is a specific choice of 'pruning implementation or configuration', which does not impact the calculation or definition of the sparsity metric (some methods prefer avoiding pruning some specific modules according to some heuristic rules). Additionally, the authors of SWAMP [7] also state that "we empirically checked that including the fully-connected layers as part of the prunable parameters does not affect the results we have obtained".
>
>
> [4]. Sidak Pal Singh and Dan Alistarh. Woodfisher: Efficient second-order approximation for neural network compression. (NeurlPS 2020)
>
> [5]. Tao Lin, Sebastian U. Stich, et al. Dynamic model pruning with feedback. (ICLR 2020)
>
> [6]. Aditya Kusupati, Vivek Ramanujan, et al. Soft threshold weight reparameterization for learnable sparsity. (ICML 2020)
>
> [7]. Moonseok Choi, Hyungi Lee, et al. Sparse weight averaging with multiple particles for iterative magnitude pruning. (ICLR 2024)
>
> -----
> **Q5. Why termed as SFPK?**
>
> **Response to Q5.** We coined the proposed analysis framework as SFPK to highlight that it is a sparsity-indexed FPK equation, in contrast to the thermodynamic time-indexed FPK equation.

---

> > ### Comment · Reviewer_3E6s · 2024-11-23
> >
> > Thank you to the authors for addressing my concerns. I truly enjoyed the additional results and was impressed (and a bit apologetic :)) by the substantial amount of work done within the limited rebuttal period. The responses to questions about multiple particles (e.g., Is the training cost reasonable? (W2); How diverse are the particles? (Q3)) and the rationality of the initial solution (e.g., How much pre-training is necessary? (W4); What happens with extremely random initialization? (Q2)) were particularly insightful.
> >
> > Also, I raised concerns about sparsity calculation (Q4) based on my negative experiences reproducing previous work, particularly regarding the impact of including or excluding the final dense layer. As you noted in your response, simply clarifying in the paper how pruning is performed and sparsity is calculated should suffice; the rest can be left to readers and future researchers.
> >
> > Regarding the high sparsity on ResNet50 with ImageNet (W5), I believe the proposed methodology achieves sufficiently competitive performance without compromising the significance of this work. I am also pleased to see that the paper now includes additional 95% sparsity results and addresses relevant recent work.
> >
> > While some results are incomplete due to time constraints (as noted in my postscript), the authors have already presented extensive additional results, and I trust they will integrate the points discussed during the rebuttal period into the final version. With the expectation that the final version will present a theoretically well-grounded methodology supported by comprehensive experimental results, I have revised my score from R6C2 to R8C3.

---

> > > ### Author Response · Authors · 2024-11-24
> > >
> > > We would like to once again express our heartfelt gratitude to you for dedicating your time and effort to reviewing our rebuttal and paper.
> > >
> > > We remain open and willing to address any further questions or concerns you may have.

---

### Meta-Review · Area_Chair_QJhj · 2024-12-19

**Metareview:**

All the reviewers are happy with the quality of the paper, and the AC is on the same page. It is a strong submission both in terms of theory and experiments. The proposed algorithm using FPK equation for probabilistic neural pruning is a novel approach, and the experimental results are thorough and convincingly support the claims.

**Additional Comments On Reviewer Discussion:**

While all the reviewers recommended acceptance, there were some initial concerns raised during the review process. However, the authors did an excellent job addressing these concerns during the rebuttal period. The consensus among the reviewers remained unchanged following the discussion phase.

---

### Decision · Program_Chairs · 2025-01-22

Accept (Spotlight)